# Navigating through the O(N) archipelago

Benoit Sirois[1, 2]

[1]*Laboratoire de Physique de l'École normale supérieure,*
*ENS, Université PSL, CNRS, Sorbonne Université,*
*Université de Paris, F-75005 Paris, France*
[2]*Institut des Hautes Études Scientifiques,*
*35 Route de Chartres, 91440 Bures-sur-Yvette, France*

(Dated: March 23, 2022)

## Abstract

A novel method for finding allowed regions in the space of CFT-data, coined navigator method, was recently proposed in [1]. Its efficacy was demonstrated in the simplest example possible, i.e. that of the mixed-correlator study of the 3D Ising Model. In this paper, we would like to show that the navigator method may also be applied to the study of the family of $d$-dimensional $O(N)$ models. We will aim to follow these models in the $(d, N)$ plane. We will see that the "sailing" from island to island can be understood in the context of the navigator as a parametric optimization problem, and we will exploit this fact to implement a simple and effective path-following algorithm. By sailing with the navigator through the $(d, N)$ plane, we will provide estimates of the scaling dimensions $(\Delta_\phi, \Delta_s, \Delta_t)$ in the entire range $(d, N) \in [3, 4] \times [1, 3]$. We will show that to our level of precision, we cannot see the non-unitary nature of the $O(N)$ models due to the fractional values of $d$ [2] or $N$ [3] in this range. We will also study the limit $N \to 1$, and see that we cannot find any solution to the unitary mixed-correlator crossing equations below $N = 1$.

**CONTENTS**

## I. INTRODUCTION

In many cases, the numerical conformal bootstrap in $d > 2$ has shown that the combination of crossing symmetry and unitarity is often sufficient in isolating theories in finite regions, or "islands", inside of the space of CFT-data (see e.g. [4–10]). Crucially, the shape and size, or even existence of these islands depend upon the assumptions one makes on the spectrum of the theory he/she is trying to isolate. These assumptions are usually extrapolated from a certain perturbative regime ($\epsilon$-expansion, large-N, etc.). Their validity in the non-perturbative regime is sometimes up in the air, and one usually has to play around with these assumptions. It is therefore natural to want to solve these theories directly in the perturbative regime, where the gap assumptions can be made robustly, and flow towards the non-perturbative regime, adjusting the gap assumptions as necessary. One might also be interested in following a CFT through some external parameter space because he/she expects something interesting to happen, for example a collision of two fixed points, at a critical value $p_c$ of the external parameter(s) $p$. Such an enterprise raises certain concerns:

- Performing a scan of the search space for each external parameter value is often times prohibitively expensive, even more so if there are many external parameters. One would benefit greatly if there was a way that both the flow in the external parameter space and in the search space could be made more efficient.

- For certain ranges of external parameters, like fractional spacetime dimensions, CFTs are expected to be non-unitary [2], although the bootstrap of the Ising Model was shown to be insensitive to this [11] (see [12–16] for other unitary bootstrap studies of non-unitary CFTs).

The goal of this paper is thus twofold: firstly, we would like to study the consequences of the non-unitary nature of another prototypical CFT, the $d$-dimensional $O(N)$ model, due to fractional

values of $d$ [2] and $N$ [3]. In the process, we will present estimates of scaling dimensions, both from the bootstrap and from the resummation of six loop $4 - \epsilon$ expansions [17, 18], for the range $(d, N) \in [3, 4] \times [1, 3]$. We will show that above $N = 1$, the bootstrap seems insensitive, to our degree of precision, to the non-unitarity due to fractional values of $d$ or $N$. $N = 1$ will represent for our analysis the absolute lower bound for $N$, since we will be unable to find a solution to the mixed-correlator crossing equations below it.

The main goal of this paper will be to lay out a method that aims to address the first concern. To follow the $O(N)$ islands through the $(d, N)$ plane, we will invoke the newly-developed navigator method of [1] (see also [19–21] for recent works using the navigator method). If $x$ denotes the set of parameters to scan over in a certain bootstrap setup, a navigator function $\mathcal{N}(x)$ is a function which is positive ouside of the allowed regions for $x$, and negative inside. One can thus flow to the allowed region(s) by minimizing the navigator function. It was shown in [1] that such a navigator can be constructed in general, and that in the example considered it correctly reproduced the 3D Ising Island. The minimum $x_{min}$ was shown to give a good estimate of the location of the true CFT. We will show that this general construction also defines a valid navigator for the case of the $O(N)$ model. Here, the navigator will also depend on some external parameters: $\mathcal{N} := \mathcal{N}(x; p)$, where $p = (d, N)$. To follow the islands through $d$ and $N$, we will trace out the curve $x_{min}(p)$. We will see that a simple formula relates derivative information at $p = p_0$ to the location of $x_{min}(p_0 + \delta p)$, and use this to help us sail through the $O(N)$ archipelago [4].

## II.  THEORY

We will aim to study the $O(N)$ model for $3 \leq d \leq 4$, starting from the perturbative regime $\epsilon = 4 - d \ll 1$. In this regime, the $O(N)$ model may be viewed as the weakly-coupled fixed point of the lagrangian

$$\mathcal{L}(\phi) = (\partial \phi)^2 + m\phi^2 + \lambda \left(\phi^2\right)^2 \tag{1}$$

for the $N$-component field $\phi_i$. The critical exponents associated to this fixed point may be given as series-expansions in the expansion parameter $\epsilon$. These can in turn be related to the scaling dimensions of certain operators in the corresponding CFT. Some operators of interest to us will be $\phi$, the two lowest-dimensional $O(N)$ singlets $s = \phi^2$ and $s' = (\phi^2)^2$, and the two lowest-dimensional two-index symmetric-traceless tensors $t = \phi_i \phi_j - \frac{1}{N}\delta_{ij}\phi^2$ and $t' = \phi^2(\phi_i \phi_j - \frac{1}{N}\delta_{ij}\phi^2)$. To the lowest non-trivial order, their dimensions are [17, 18, 22]

$$\Delta_\phi = \frac{d-2}{2} + \frac{\eta(\epsilon)}{2} = 1 - \frac{\epsilon}{2} + \frac{N+2}{4(N+8)^2}\epsilon^2 + \mathcal{O}(\epsilon^3)$$

$$\Delta_s = d - \nu^{-1}(\epsilon) = 2 - \frac{6}{N+8}\epsilon + \mathcal{O}(\epsilon^2)$$

$$\Delta_t = d - d_f(\epsilon) = 2 - \frac{N+6}{N+8}\epsilon + \mathcal{O}(\epsilon^2) \tag{2}$$

$$\Delta_{s'} = d + \omega(\epsilon) = 4 + \mathcal{O}(\epsilon^2)$$

$$\Delta_{t'} = d - y_{4,2}(\epsilon) = 4 - \frac{N}{N+8}\epsilon + \mathcal{O}(\epsilon^2) \quad .$$

6-loop expansions for the critical exponents $\eta(\epsilon)$, $\nu^{-1}(\epsilon)$ and the correction-to-scaling exponent $\omega(\epsilon)$ can be found in [18] ($\eta(\epsilon)$ is actually known to 8-loops [23], and $\nu^{-1}(\epsilon)$ to 7 [24]). The fractal

dimension $d_f(\epsilon)$ is also known to 6-loops [17], while the RG dimension $y_{4,2}(\epsilon)$ is known to only 5-loops [22]. A collection of all known $\epsilon$-expansion CFT data was recently given in [25], and we will refer to it for data not listed in Eq. (2).

We will want to compare bootstrap results to the epsilon expansions presented above. The epsilon expansion for critical exponents is well-known to give divergent series. A resummation procedure is necessary in order to give meaningful results at finite $\epsilon$. We will resum these series using the algorithm of Borel-Leroy transform with conformal mapping laid out in Section V of [18]. This algorithm is quite elaborate, and involves many parameters. We use the same values for these parameters as those cited in [18], and refer the reader to this paper for its description.

Something, although much less than for scaling dimensions, is also known about OPE coefficients in the $\epsilon$-expansion. A quantity we will need in the future is the ratio of OPE coefficients $\theta(\epsilon) = \arctan \frac{\lambda_{sss}(\epsilon)}{\lambda_{\phi\phi s}(\epsilon)}$, which is known to $\mathcal{O}(\epsilon)$ [25–27] [1]:

$$\theta(\epsilon) = \arctan 2 - \frac{2(N+2)}{5(N+8)}\epsilon + \mathcal{O}(\epsilon^2) \quad . \tag{3}$$

## III.   SETUP AND FIRST EXAMPLE

We will be considering in this work the crossing equation arising from the 4-point functions $\langle\phi\phi\phi\phi\rangle$, $\langle\phi\phi ss\rangle$ and $\langle ssss\rangle$, as was first considered in [4]. As mentioned in the previous section, $\phi_i$ and $s$ are the lowest-lying scalar operators transforming in the vector and singlet representations of $O(N)$ respectively. The crossing equation deriving from this set of mixed correlators was shown to be [4]

$$\sum_{\mathcal{O}\in S}\begin{pmatrix}\lambda_{\phi\phi\mathcal{O}} & \lambda_{ss\mathcal{O}}\end{pmatrix}\vec{V}_{S,\Delta,\ell}\begin{pmatrix}\lambda_{\phi\phi\mathcal{O}}\\\lambda_{ss\mathcal{O}}\end{pmatrix} + \sum_{R\in\{A,T\}}\sum_{\mathcal{O}\in R}\lambda^2_{\phi\phi\mathcal{O}}\vec{V}_{R,\Delta,\ell} + \sum_{\mathcal{O}\in V}\lambda^2_{\phi s\mathcal{O}}\vec{V}_{V,\Delta,\ell} = 0 \quad , \tag{4}$$

$$\vec{V}_{V,\Delta,\ell} = \begin{pmatrix}0\\0\\0\\0\\F^{\phi s,\phi s}_{-,\Delta,\ell}\\(-1)^\ell F^{s\phi,\phi s}_{-,\Delta,\ell}\\-(-1)^\ell F^{s\phi,\phi s}_{+,\Delta,\ell}\end{pmatrix}, \quad \vec{V}_{T,\Delta,\ell} = \begin{pmatrix}F^{\phi\phi,\phi\phi}_{-,\Delta,\ell}\\\left(1-\frac{2}{N}\right)F^{\phi\phi,\phi\phi}_{-,\Delta,\ell}\\-\left(1+\frac{2}{N}\right)F^{\phi\phi,\phi\phi}_{+,\Delta,\ell}\\0\\0\\0\\0\end{pmatrix}, \quad \vec{V}_{A,\Delta,\ell} = \begin{pmatrix}F^{\phi\phi,\phi\phi}_{-,\Delta,\ell}\\-F^{\phi\phi,\phi\phi}_{-,\Delta,\ell}\\F^{\phi\phi,\phi\phi}_{+,\Delta,\ell}\\0\\0\\0\\0\end{pmatrix},$$

---

[1] We thank Johan Henriksson for pointing us to the result for $\lambda_{sss}(\epsilon)$.

$$\vec{V}_{S,\Delta,\ell} = \begin{pmatrix} \begin{pmatrix} 0 & 0 \\ 0 & 0 \end{pmatrix} \\ \begin{pmatrix} F_{-,\Delta,\ell}^{\phi\phi,\phi\phi} & 0 \\ 0 & 0 \end{pmatrix} \\ \begin{pmatrix} F_{+,\Delta,\ell}^{\phi\phi,\phi\phi} & 0 \\ 0 & 0 \end{pmatrix} \\ \begin{pmatrix} 0 & 0 \\ 0 & F_{-,\Delta,\ell}^{ss,ss} \end{pmatrix} \\ \begin{pmatrix} 0 & 0 \\ 0 & 0 \end{pmatrix} \\ \begin{pmatrix} 0 & \frac{1}{2}F_{-,\Delta,\ell}^{\phi\phi,ss} \\ \frac{1}{2}F_{-,\Delta,\ell}^{\phi\phi,ss} & 0 \end{pmatrix} \\ \begin{pmatrix} 0 & \frac{1}{2}F_{+,\Delta,\ell}^{\phi\phi,ss} \\ \frac{1}{2}F_{+,\Delta,\ell}^{\phi\phi,ss} & 0 \end{pmatrix} \end{pmatrix} . \tag{5}$$

We have suppressed above the dependence of the convolved blocks $F_{\pm,\Delta,\ell}^{ij,kl}(u,v)$ on the cross-ratios $(u,v)$. For their exact expression, see e.g. (2.4) of [4]. $S, T, A, V$ refer respectively to the singlet, two-index traceless-symmetric, two-index antisymmetric and vector representations of $O(N)$ which are present in either the $\phi_i \times \phi_j$ OPE $(S, T, A)$, the $\phi_i \times s$ OPE $(V)$ or the $s \times s$ OPE $(S)$. Say we want to constrain some set of CFT data $x$ by making certain assumptions on the spectrum of the $O(N)$ models, and checking if these assumptions at our parameters $x$ allow a solution to Eq. (4). [1] describes a simple recipe to define a *GFF navigator function* $\mathcal{N}^{\mathrm{GFF}}(x)$ (we will omit the superscript GFF in the rest of the paper) which is positive when $x$ is disallowed by Eq. (4) under the assumptions we made, and negative when $x$ is allowed: we can add to Eq. (4) a contribution $\lambda \vec{M}_{GFF}$ corresponding to the operators whose dimensions are below the gaps we assumed in a solution where $\phi_i$ is an $O(N)$ generalized free field (GFF) and where $s$ is another independent GFF, and minimize $\lambda$ [2]. The OPEs for this $O(N)$ mixed-correlator GFF solution are

$$\phi_i \times \phi_j = \delta_{ij} \sum_{\ell\,\mathrm{even}} \sum_{n \in \mathbb{Z}_{\geq 0}} \lambda_{\phi\phi(n\ell)}^{\mathrm{S}} \text{`` } \phi_k \Box^n \partial^\ell \phi_k \text{ ''} + \sum_{\ell\,\mathrm{even}} \sum_{n \in \mathbb{Z}_{\geq 0}} \lambda_{\phi\phi(n\ell)}^{\mathrm{T}} \text{`` } \phi_{(i} \Box^n \partial^\ell \phi_{j)} \text{ ''} +$$

$$\sum_{\ell\,\mathrm{odd}} \sum_{n \in \mathbb{Z}_{\geq 0}} \lambda_{\phi\phi(n\ell)}^{\mathrm{A}} \text{`` } \phi_{[i} \Box^n \partial^\ell \phi_{j]} \text{ ''}$$

$$\phi_i \times s = \sum_{\ell} \sum_{n \in \mathbb{Z}_{\geq 0}} \lambda_{\phi s(n\ell)}^{\mathrm{V}} \text{`` } \phi_i \Box^n \partial^\ell s \text{ ''} \tag{6}$$

---

[2] Another form of navigator function, coined the $\Sigma$-*navigator*, was proposed in [1], which amounts to adding a different contribution to the crossing equation that still ensures the augmented crossing equation always has a solution.

$$s \times s = \sum_{\ell \text{ even}} \sum_{n \in \mathbb{Z}_{\geq 0}} \lambda^{\text{S}}_{ss(n\ell)} \text{`` } s\Box^n \partial^\ell s \text{ ''} \quad ,$$

where the various dimensions and OPE coefficients are given by [28, 29]

$$\Delta_{\text{``}A\Box^n \partial^\ell B\text{''}} = \Delta_A + \Delta_B + 2n + \ell \tag{7}$$

$$\left(\lambda^{\text{T}}_{\phi\phi(n\ell)}\right)^2 = \left(\lambda^{\text{A}}_{\phi\phi(n\ell)}\right)^2 = c_{n,\ell}\left(\Delta_\phi, \Delta_\phi\right)$$
$$\left(\lambda^{\text{S}}_{\phi\phi(n\ell)}\right)^2 = \frac{2}{N} c_{n,\ell}\left(\Delta_\phi, \Delta_\phi\right)$$
$$\left(\lambda^{\text{S}}_{ss(n\ell)}\right)^2 = 2\, c_{n,\ell}\left(\Delta_\phi, \Delta_\phi\right) \tag{8}$$
$$\left(\lambda^{\text{V}}_{\phi s(n\ell)}\right)^2 = c_{n,\ell}\left(\Delta_\phi, \Delta_s\right) \quad ,$$

$$c_{n,\ell}\left(\Delta_1, \Delta_2\right) = \frac{2^\ell (\Delta_1 - \frac{d-2}{2})_n (\Delta_2 - \frac{d-2}{2})_n}{\ell! n! (\frac{d-2}{2} + \ell + 1)_n (\Delta_1 + \Delta_2 + n - d + 1)_n} \times$$
$$\frac{(\Delta_1)_{\ell+n} (\Delta_2)_{\ell+n}}{(\Delta_1 + \Delta_2 + 2n + \ell - 1)_\ell (\Delta_1 + \Delta_2 + n + \ell - \frac{d}{2})_n} \tag{9}$$

with $(\cdot)_x$ the Pochhammer symbol and the differences between Eq. (9) and its equivalents in [28, 29] are due to different conformal block normalizations.

For the majority of the paper, we will take the parameters on which the navigator depends to be $x = (\Delta_\phi, \Delta_s, \Delta_t)$. We will assume the existence of a spin-1 conserved current $J_\mu$ in the antisymmetric representation with dimension $\Delta_{J^\mu} = d-1$, and the existence of a spin-2, dimension $d$ singlet operator corresponding to the stress-energy tensor $T_{\mu\nu}$. We will assume some gaps above $\phi, s, t, J_\mu$ and $T_{\mu\nu}$, and impose that the rest of the spectrum respects unitarity bounds. Finally, we will impose a twist gap $\tau = 10^{-10}$ above the unitarity bounds to try to forbid solutions where spurious operators would sit exactly at the unitarity bounds. Computing the navigator function then amounts to solving the following optimization problem:

$$\mathcal{N}(x) = \max_{\vec{\alpha}} \quad \begin{pmatrix} 1 & 1 \end{pmatrix} \vec{\alpha} \cdot \vec{V}_{S,0,0} \begin{pmatrix} 1 \\ 1 \end{pmatrix} \qquad \text{such that} \tag{10}$$

$$\vec{\alpha} \cdot \vec{M}_{GFF} = -1$$

$$\vec{\alpha} \cdot \left(\vec{V}_{S,\Delta_s,0} + \vec{V}_{V,\Delta_\phi,0} \otimes \begin{pmatrix} 1 & 0 \\ 0 & 0 \end{pmatrix}\right) \succcurlyeq 0$$

$$\vec{\alpha} \cdot \vec{V}_{T,\Delta_t,0} \geq 0$$
$$\vec{\alpha} \cdot \vec{V}_{A,d-1,1} \geq 0$$
$$\vec{\alpha} \cdot \vec{V}_{S,d,2} \succcurlyeq 0$$

$$\vec{\alpha} \cdot \vec{V}_{V,\Delta,\ell} \geq 0 \quad \begin{cases} \Delta \geq \Delta_\phi^* & \ell = 0 \\ \Delta \geq d + \ell - 2 + \tau & \ell > 0 \end{cases}$$

$$\vec{\alpha} \cdot \vec{V}_{S,\Delta,\ell} \succcurlyeq 0 \quad \begin{cases} \Delta \geq \Delta_s^* & \ell = 0 \\ \Delta \geq \Delta_{T_{\mu\nu}}^* & \ell = 2 \\ \Delta \geq d + \ell - 2 + \tau & \ell > 2 \end{cases}$$

$$\vec{\alpha} \cdot \vec{V}_{T,\Delta,\ell} \geq 0 \quad \begin{cases} \Delta \geq \Delta_t^* & \ell = 0 \\ \Delta \geq d + \ell - 2 + \tau & \ell > 0 \end{cases}$$

$$\vec{\alpha} \cdot \vec{V}_{A,\Delta,\ell} \geq 0 \quad \begin{cases} \Delta \geq \Delta_{J_\mu}^* & \ell = 1 \\ \Delta \geq d + \ell - 2 + \tau & \ell > 1 \end{cases}$$

An example of a valid $\vec{M}_{GFF}$ will be given shortly. It is well known by now that such a bootstrap problem can be recast in the form of a semidefinite program (SDP), which we solve with SDPB 2.4.0 [30, 31]. We compute conformal blocks with scalar_blocks [32], and use simpleboot [33] as our user-interface in setting up the bootstrap problem. The numerical parameter which governs the size of allowed regions is the derivative order $\Lambda$ to which we Taylor expand the functionals in $\vec{\alpha}$. Unless stated otherwise, we will use $\Lambda = 19$. As discussed in [1], the navigator function defined in Eq. (10) is generically concave far enough away from allowed regions, and asymptotes to a constant $\mathcal{N}_{max}$. In our numerical implementation, we will, just as in [1], decide to work with the transformed navigator

$$f(x) = \frac{\mathcal{N}(x)}{\mathcal{N}_{max} - \mathcal{N}(x)} \quad . \tag{11}$$

We had found in [1] that this improves the efficacy of the navigator method far away from the allowed regions. As defined here, $f(x)$ is positive if and only if the actual navigator $\mathcal{N}(x)$ is positive.

We expect that the assumptions laid in Eq. (10) with judicious choices of gaps $\vec{\Delta}^* = (\Delta_\phi^*, \Delta_s^*, \Delta_t^*, \Delta_{J_\mu}^*, \Delta_{T_{\mu\nu}}^*)$ should lead to small isolated islands in the three-dimensional parameter space where $x$ lives for every value of $(d, N)$ we will consider. Starting from an initial guess $x_0$, we will sail to these isolated islands by minimizing the transformed function Eq. (11) using the quasi-Newton BFGS algorithm laid out in Section 5.2 of [1]. This algorithm uses gradient information. It therefore greatly benefits from the fact that the gradient of the solution to the SDP associated to Eq. (10) may be computed "for free", i.e. only knowing the solution of the SDP at the point where the gradient is demanded, and the variations of the SDP. The exact formula for this gradient is given in Section 4.2 of [1].

In the following example, we will use the navigator to sail into the $d = 3$ $O(2)$ island. We will use as our starting point $x_0 = (0.519, 1.5051, 1.2358)$, which corresponds to the resummed values of the six-loop expansions of the dimensions expanded to lower order in Eq. (2). We will set $\vec{\Delta}^* = (3, 3, 3, 2.5, 3.5)$. The first three gaps are those used in [4], and amount to stating that $\phi$, $s$ and $t$ are respectively the only relevant vector, singlet and traceless-symmetric scalars in the theory. We use a very conservative gap of $\frac{1}{2}$ above the conserved current and stress-energy tensor. The BFGS algorithm requires a bounding box $\mathfrak{B} = [\Delta_\phi^{\min}, \Delta_\phi^{\max}] \times [\Delta_s^{\min}, \Delta_s^{\max}] \times [\Delta_t^{\min}, \Delta_t^{\max}]$ inside of which the search will take place. We choose the very conservative $\mathfrak{B} = [0.517, 0.522] \times$

$[1.45, 1.55] \times [1.2, 1.3]$ based on a previous bootstrap study with a nearly identical setup (see Fig. 4 of [4]). This bounding box has two purposes: it both prohibits BFGS from flowing into allowed regions which have nothing to do with the $O(N)$ models (referred in the bootstrap jargon as the "peninsulas") and provides a scale for the initial hessian $B_0$. Given the bounding box $\mathfrak{B}$, we perform the change of variables $x \to y(x)$ defined by

$$x_i = x_{0;i} + (\Delta_{x_i}^{\max} - \Delta_{x_i}^{\min}) \cdot r \cdot y_i \quad . \tag{12}$$

Then when an approximate Hessian is not otherwise known, we use

$$B_0 = \left\| \nabla \tilde{f}(0) \right\| \mathbb{1} \quad , \tag{13}$$

where $\tilde{f}(y) = f(x)$, as the initial Hessian. This is completely equivalent to the prescription for the initial Hessian given in Algortihm 1 of [1]. $r$ of Eq. (12) gives the ratio of the bounding box that we wish to explore in each direction in the first step. It was set to 0.2 in [1]. We will throughout this paper use the smaller $r = 0.05$ as we will usually stay close to the islands we're looking for, and won't need to explore too much of the parameter space. BFGS terminates once the maximum norm ($\|a\|_\infty = \max(|a_1|, |a_2|, \ldots)$) of the gradient of the transformed function $\tilde{f}$ reaches a cutoff $g_{\text{tol}}$, where we chose $g_{\text{tol}} = 10^{-8}$ [3]. This is the standard termination criteria of the SciPy [34] implementation of BFGS we were using. Finally, to actually compute $\mathcal{N}(x)$, we have to define a valid GFF contribution vector $\vec{M}_{GFF}$. If we defined it at every $x$ as the contribution from all GFF vectors with dimensions below our gaps $\vec{\Delta}^*$, we would run the risk that it might change as $\Delta_\phi$ and $\Delta_s$ vary within a BFGS run, resulting in a discontinuous change of the navigator function (remember that the dimensions of those GFF operators are given by Eq. (7)). To make sure that we have at least all the operators we need (and maybe some superfluous ones) throughout the BFGS run, we choose $\vec{M}_{GFF}$ to be the contribution from the minimal set of operators required for $(\Delta_\phi, \Delta_s) = (\Delta_\phi^{\min}, \Delta_s^{\min})$. With our choice of $\vec{\Delta}^*$, this means we take

$$\vec{M}_{GFF} = \frac{1}{2} c_{0,0} (\Delta_\phi, \Delta_\phi) \begin{pmatrix} 1 & 0 \end{pmatrix} \vec{V}_{S,2\Delta_\phi,0} \begin{pmatrix} 1 \\ 0 \end{pmatrix} + c_{0,0} (\Delta_s, \Delta_s) \begin{pmatrix} 0 & 1 \end{pmatrix} \vec{V}_{S,2\Delta_s,0} \begin{pmatrix} 0 \\ 1 \end{pmatrix} +$$
$$\frac{1}{2} c_{0,2} (\Delta_\phi, \Delta_\phi) \begin{pmatrix} 1 & 0 \end{pmatrix} \vec{V}_{S,2\Delta_\phi+2,2} \begin{pmatrix} 1 \\ 0 \end{pmatrix} + \tag{14}$$
$$\frac{1}{2} \left( c_{0,0} (\Delta_\phi, \Delta_\phi) \vec{V}_{T,2\Delta_\phi,0} + c_{0,1} (\Delta_\phi, \Delta_\phi) \vec{V}_{A,2\Delta_\phi+1,1} + c_{0,0} (\Delta_\phi, \Delta_s) \vec{V}_{V,\Delta_\phi+\Delta_s,0} \right) \quad .$$

We of course could have multiplied $\vec{M}_{GFF}$ with any positive prefactor. This choice would influence the value of $\mathcal{N}_{max}$, and with the one made here, $\mathcal{N}_{max} = 2$. We present in Fig. 1 the results of this example BFGS run. It took 34 total function calls and 24 BFGS iterations (which are function calls accepted in the line searches done by BFGS) to reach the minimum of the transformed navigator. The minimum is reached at $x_{min} = (0.518899, 1.50739, 1.23446)$. Because our assumptions were very close to those of [4], it is no surprise that this allowed value is consistent with the $3D$ allowed region of Fig. 4 of [4].

We may know the size of the island by computing the maximal and minimal allowed values for

---

[3] The change of variables Eq. (12) obviously impacts when this termination criterion is reached.

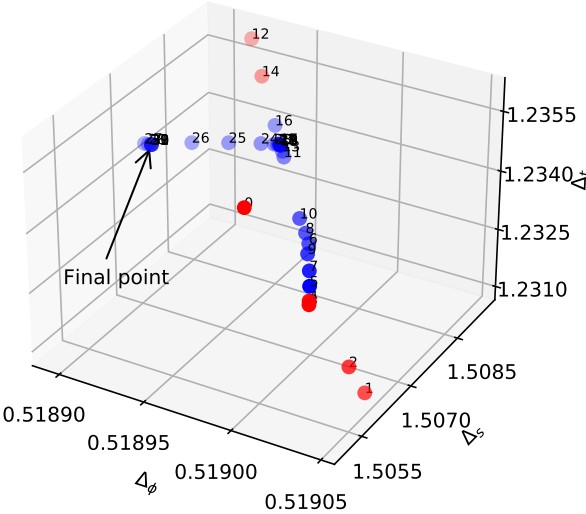

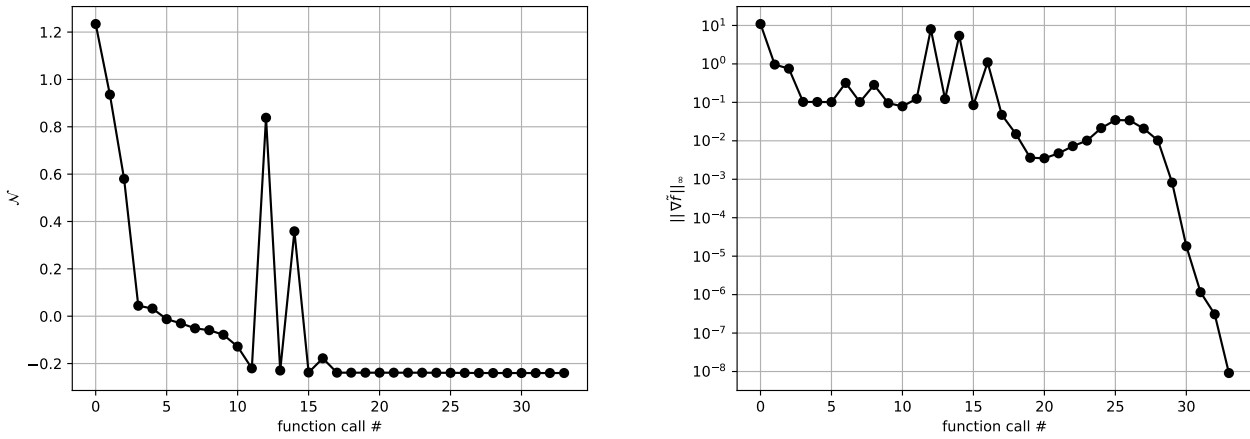

FIG. 1.    Top: BFGS run for the example described in Section III at derivative order $\Lambda = 19$. Points are numbered, starting from 0, by their function call number (and not the BFGS step number). Allowed points are marked in blue, and disallowed in red. Bottom left: Value of the navigator function at every function call. We can see it takes 5 function calls to reach the island. Bottom right: Maximum norm of the gradient, in the coordinates of Eq. (12), of the transformed function Eq. (11). BFGS terminates once the norm goes below gtol $= 10^{-8}$.

each coordinate in $x$ with the "Constrained BFGS" algorithm of Section 6 of [1]. This algorithm attempts to extremize $x$ along some given direction by performing a sequence of line searches whose directions are informed by an approximate quadratic model of the navigator $\mathcal{N}(x)$ which is updated after every line search. These line searches are forced to remain close to the boundary of the island, and the algorithm terminates once $|\mathcal{N}(x)|$ or the component of the gradient of $\mathcal{N}(x)$ perpendicular to the extremization direction go below some tolerance $g_{\text{tol}}$. We chose for this

example $g_{\text{tol}} = 10^{-10}$. The algorithm also requires a starting point inside of the island and an initial guess for the Hessian. We chose for this the minimum $x_{min}$ of the BFGS run in Fig. 1 and the approximate Hessian supplied by BFGS at this minimum point. Fig. 2 shows the path taken for the minimization of $\Delta_\phi$, one of the six extremization runs. The resulting bounds from all six runs are

$$x_{\text{allowed}} \in [0.518344\ldots, 0.520557\ldots] \times [1.49996\ldots, 1.51978\ldots] \times$$
$$[1.23091\ldots, 1.24189\ldots] \tag{15}$$

Comparison to Figure 4 of [4] (for which the assumptions, bar the existence of a stress-tensor and conserved current, were the same as those used here) suggests that the region of negative $O(N)$ navigator does reproduce the actual allowed region, as was the case for the Ising model [1, 19] (see also [21] for another application of the navigator method, this time to the $\mathcal{N} = 1$ super-Ising model). It is interesting to note, by comparing to Figure 3 of [4], that the addition of the assumptions of a single relevant traceless-symmetric scalar, of a stress-tensor and of a conserved current seemed only to carve out a small portion at the upper-right of the island in the $(\Delta_\phi, \Delta_s)$ plane (such a behaviour was already discussed in [35] for only the addition of the stress-tensor).

We have demonstrated in this section that the GFF navigator construction may be applied successfully, as expected, to the case of the $O(N)$ model, showing agreement between allowed regions obtained with the navigator method and allowed regions obtained with the usual "binary" allowed/disallowed bootstrap. With the help of a small trick described in the following section, we will use this construction in Sections V and VI to sail through the $(d, N)$ plane.

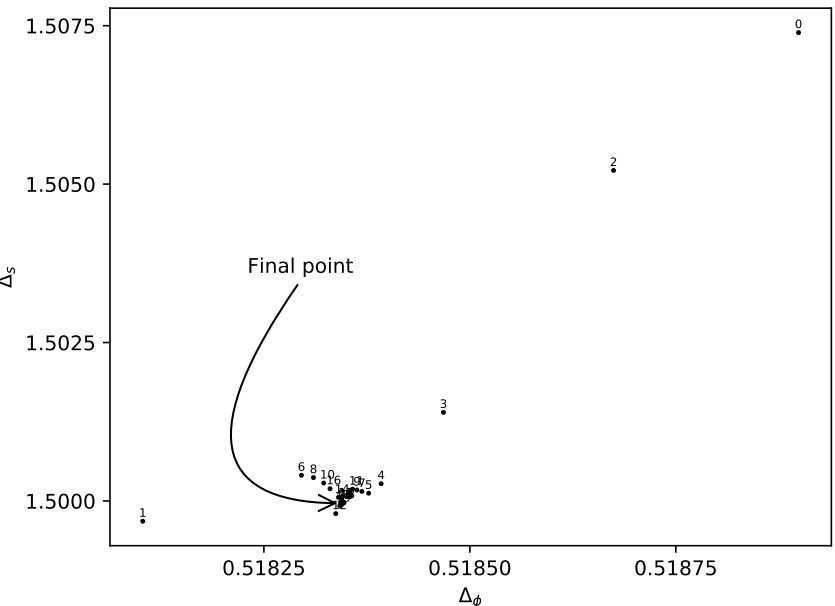

FIG. 2. $(\Delta_\phi, \Delta_s)$ projection of the "Constrained BFGS" run for the minimization of $\Delta_\phi$ at $\Lambda = 19$. The run terminates on the tip of the arrow at $x_f = (0.518344, 1.49996, 1.23104)$. The "Constrained BFGS" algorithm allows for the probing of disallowed points in its search for extremal allowed values; e.g. point 1 here has a navigator of 1.22824.

## IV.   SIMPLE PATHFOLLOWING

We will, in the following sections, attempt to follow the $O(N)$ models through the space of our external parameters $p = (d, N)$. As argued in Section III, we will do so by following the minimum $x_{min}(p)$ of the GFF navigator function. To move as fast as possible in the external parameter space, we need to establish a way to use the knowledge of a solution at a certain $p$ to sail to a solution at a nearby parameter $p + \delta p$. Because each $x_{min}(p)$ is the solution to a minimization problem, the problem we are faced with is referred to in the optimization literature as "parameteric optimization", and what we are trying to do, as "pathfollowing" (see [36, 37] for pedagological references on this subject). We will implement the following simple pathfollowing method. Say we know the minimum $x_0$ for some $p = p_0$. Then, under certain conditions on the function being minimized, $x_{min}$ is given in some neighbourhood of $x_0$ as a function of $p$: $x_{min} = \tilde{x}(p)$ with $\tilde{x}(p_0) = x_0$, and the first order variation of the position of the minimum is given (in Cartesian coordinates) at $x_0$ by (see [36], Theorem 4.1)

$$\frac{\partial \tilde{x}_i}{\partial p^n}(p_0) = -\left(\mathrm{B}_f^{-1}(x_0; p_0)\right)_{ij} \frac{\partial^2 f}{\partial \tilde{x}^j \partial p^n}(x_0; p_0) \quad , \tag{16}$$

with $f$ the function from Eq. (11) which we minimize, and the Hessian given by $(\mathrm{B}_f(x, p))_{ij} = \frac{\partial^2 f}{\partial \tilde{x}^i \partial \tilde{x}^j}(x; p)$. This of course suggests the following: somehow sail to the first desired minimum (e.g. we will see in Section V that this can be done quite efficiently in a perturbative regime starting from known field-theory results). At this minimum, the Hessian and the mixed second derivative may be computed either by finite differences or by using the quadratic variation formula of Appendix C of [1] (we have found that the approximate Hessian obtained at the end of the previous BFGS run is not precise enough, hence why we advise to compute it independently). Then take steps $\delta p_1, \delta p_2, \ldots$ in the external parameter space small enough so that the first order variation (16) repeatedly gives good estimates of the location of the minimum of the rescaled navigator at the new parameter values $p + \delta p_1, p + \delta p_1 + \delta p_2, \ldots$ If these steps are indeed taken small enough, we should hope that the BFGS runs for the second, third, etc. parameter values would be much shorter than the first. Because we desire to compare specific external parameter values to results obtained from other methods, we will decide to choose the step sizes by hand. If one wanted to trace out the minimum curve as efficiently as possible, the choice of step size could be optimized: see [38], Chap. 6.

Let us see how this works in practice. We will start from the solution $x_0 = x_{min}(d = 3, N = 2)$ obtained in the example of Section III, and attempt to reach solutions in $d = 3$ at nearby values of $N$. Fig. 3 shows that the gradient of the position of the minimum is indeed reproduced by Eq. (16). Using the minima predicted by (16) leads to an appreciable speedup of subsequent BFGS runs. Starting the $N = 1.9$ run at the extrapolated minimum, using the Hessian at the $N = 2$ minimum as the initial guess for the Hessian for the $N = 1.9$ run, it only takes 8 function calls to reach the true minimum. For comparison, it took 45 function calls to reach the same minimum starting from the $N = 2$ minimum. This amounts to a more than fivefold speedup. As stated in the previous paragraph, there should always exist some "optimal" step size if one wants to draw the solution curve as efficiently as possible. We have not explored ways of choosing this step automatically, but we can comment that in the example considered here, $\delta N = 0.1$ is a reasonable value as smaller step sizes don't seem to lead to an increase in efficiency that offsets the need for more steps. Indeed, this can be observed in Fig. 4, where we present the amount of

function calls taken to reach the minimum for the different steps taken in Fig. 3 (for clarity, these steps are $\delta N \in \{\pm 0.01, \pm 0.025, \pm 0.05, \pm 0.1\}$). Now armed with this pathfollowing prescription, we should be in a good position to use the navigator to sail through the $(d, N)$ plane.

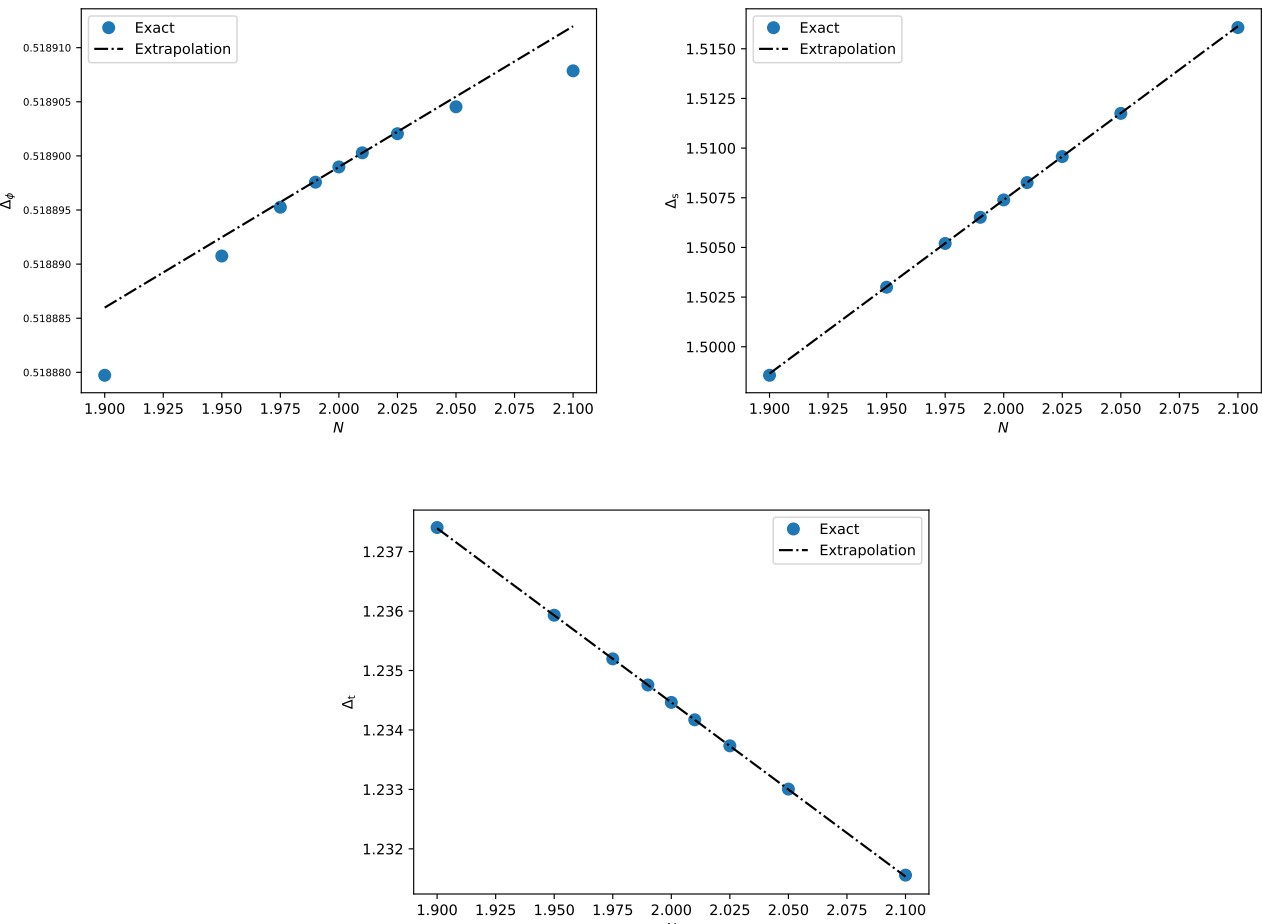

FIG. 3. Comparison of the first-order prediction (16) for the location of the minima nearby the $(d = 3, N = 2)$ minimum to their actual value. Upper left: Comparison for $\Delta_\phi$. Upper right: Comparison for $\Delta_s$. Lower middle: Comparison for $\Delta_t$.

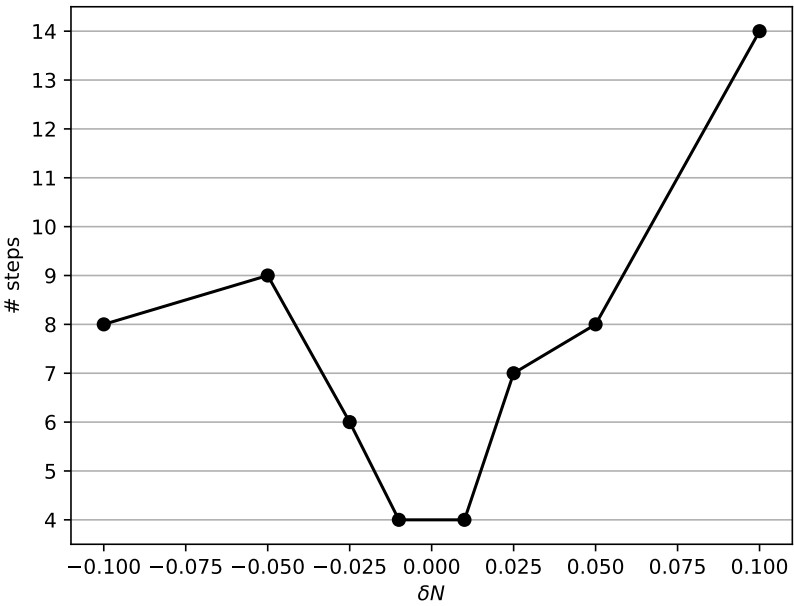

FIG. 4. Number of function calls until BFGS terminates, for the step sizes considered in Fig. 3 with initial point given by Eq. (16).

## V. SAILING THROUGH D: FROM THE PERTURBATIVE TO THE NON-PERTURBATIVE REGIME

We would now like to use the navigator construction laid out in Sections III and IV to follow the $O(N)$ models above $d = 3$, and in particular to look for signs of non-unitarity in the fractional-$d$ bootstrap solutions. Our goal will be to start from rough estimates in the perturbative regime $\epsilon = 4 - d \ll 1$, and iteratively decrease $d$, following the islands from $d = 4$ to $d = 3$ using the path-following prescription of Section IV (here with the varying external parameter $p$ being the dimension $d$). The case of $N = 1$ for $2 < d < 4$ was already considered in [11] (see also [39]). It was found that the family of kinks in the $\mathbb{Z}_2$-symmetric single-correlator $(\Delta_\sigma, \Delta_\epsilon)$-bound predicted by the $4 - \epsilon$ expansion survived all the way to $d = 2$, and that the scaling dimensions extracted from this family of kinks were in accord with those computed with the $\epsilon$-expansion in [40]. This established that the effects of the expected non-unitarity of the Ising model in fractional dimensions [2] were small enough to be inconsequential to the numerical bootstrap. In this section, we would like to elaborate on this result, and show that the numerical bootstrap appears insensitive, to our degree of precision, to the non-unitary nature of the fractional-$d$ $O(N)$ models for $N = 2, 3$. Critical exponents for the fractional-$d$ $O(N)$ models have previously been estimated using the functional RG [41] and field-theory [42–44]. Because most of the field-theory results are quite old, as stated in Section II, we will when needed resum the $\epsilon$-expansions of [17, 18] with the algorithm of [18] to use as a basis to compare bootstrap results. We use throughout this chapter as gap assumptions $\vec{\Delta}^* = (d, d, d, d - 0.5, d + 0.5)$, again looking for solutions with only one relevant scalar $O(N)$ vector, singlet and traceless-symmetric operator. We of course have to make sure that these

assumptions are never violated, which we will do at the end of this section.

As already noticed in [45] for the case of the Ising model, bootstrap islands tend to decrease significantly in size as $d$ gets closer to 4. One might fear that the navigator method would struggle to find such small islands, even more so because good initial guesses for CFT-data (e.g. $\epsilon$-expansions to high loop-orders) can be rare, and especially rare for OPE coefficients. To prove that the navigator method can handle all of these concerns, let us consider a slightly more constraining setup than (10): we want our navigator to now also depend on the OPE angle $\theta = \arctan\frac{\lambda_{sss}}{\lambda_{\phi\phi s}}$. This means we are changing the second condition of (10) to

$$\begin{pmatrix}\cos\theta & \sin\theta\end{pmatrix} \vec{\alpha} \cdot \left(\vec{V}_{S,\Delta_s,0} + \vec{V}_{V,\Delta_\phi,0} \otimes \begin{pmatrix}1 & 0\\ 0 & 0\end{pmatrix}\right) \begin{pmatrix}\cos\theta\\ \sin\theta\end{pmatrix} \geq 0 \tag{17}$$

For $N = 2$ and $\epsilon = 0.2$, we minimized (11) over the 4-parameter search space $x = (\Delta_\phi, \Delta_s, \Delta_t, \theta)$, with the initial guess for $\theta$ its free theory value $\theta = \arctan 2$, and the lowest non-trivial order estimates of Eq. (2) for the dimensions. We found that even with these very rough starting points, BFGS was eventually able to reach the allowed region, taking 81 function calls to find the first allowed point $x = (0.900470379, 1.8847650, 1.8419186, 1.0753019)$. We show a 3D projection of this BFGS run in Fig. 5, where the tiny allowed region on the right figure lies on the tip of the arrow on the left figure. For the rest of this section, including in Fig. 5, we will substitute for the scaling dimensions the more natural anomalous dimensions $(\gamma_\phi, \gamma_s, \gamma_t) = (\Delta_\phi, \Delta_s, \Delta_t) - (\frac{d-2}{2}, d-2, d-2)$.

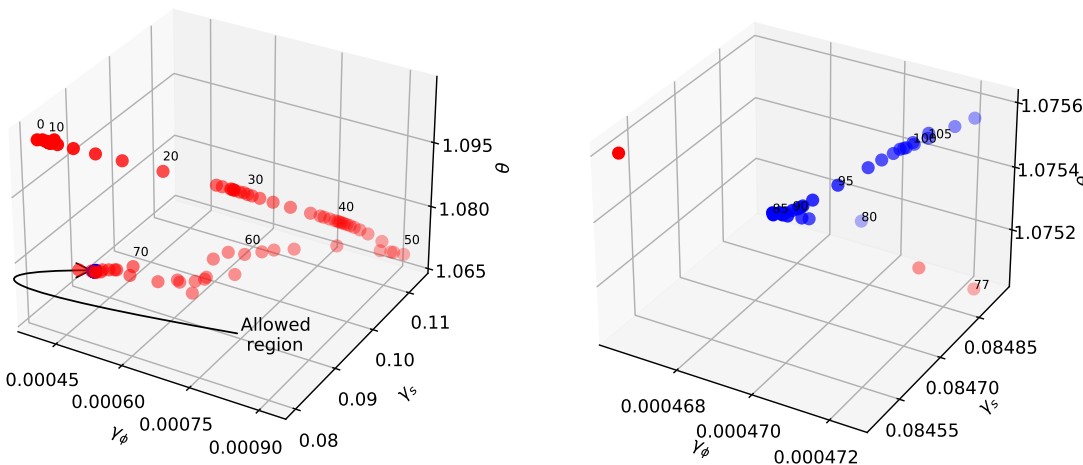

FIG. 5. 3D projection of the BFGS run at $\Lambda = 19$ for the 4-parameter setup where $x = (\Delta_\phi, \Delta_s, \Delta_t, \theta)$. Red points are disallowed, blue points are allowed. Left: Full run. The allowed region is located at the tip of the arrow and is barely visible. Points are labelled by their function call number. Right: Zoom in on the allowed region.

Fig. 5 clearly demonstrates the power of the navigator method: the very tiny island could be located in a relatively small number of steps, starting from quite a rough estimate (notice especially the different scales for the $\theta$ variable in the two plots: the island in this direction is roughly two orders-of-magnitude smaller than its distance to the starting point). With the amount of information used here, a scan would have needed to be extremely fine in the $\theta$ direction to locate an allowed

point. A rough estimate gives $10 \times 10 \times 10 \times 100 = 100{,}000$ points needed to be tested in order to find the island.

Let's now go back to the original setup of (10) and go down in dimension, from $d = 4$ to $d = 3$ in steps of $\delta d = 0.1$, for $N = 2, 3$. At small values of $\epsilon$, we find that the islands are so small that using the location of the BFGS minimum at the previous $d$ is not sufficient for the run at $d - 0.1$ to converge. We see in Fig. 6 that for $N = 2$, if one was to start at $d = 3.7$ from the minimum obtained at $d = 3.8$, BFGS would run towards the peninsula to the right of the island. With the initial guess provided by Eq. (16), as evidenced in Fig. 7, the run at $d = 3.7$ reaches the first allowed point $x = (0.00110929, 0.128838, 0.0633290)$ inside the $O(2)$ island after 64 function calls. A prescription like that of Section IV was therefore necessary in making the navigator method viable in this context. However, the initial point $x_0 = (0.000975890, 0.129421, 0.0637559)$ resulted in a navigator of $\mathcal{N}(x_0) = 1.99661$, indicating that this point was deep in the disallowed region. This is much different to the example given in Section IV, and the difference is attributable to the small of size of the small $\epsilon$ islands. Because of this, we had to make no assumption about the initial Hessian (using the Hessian at the previous minimum would be a bad guess if the starting point is not close enough to the next minimum), which partly explains why it took a considerable amount of function calls to even reach the island.

The outcome of the full path-following for $N = 2, 3$ is presented in Table I. For greater clarity, we also plot these results for each of the 3 parameters in Fig. 8. We observe clear agreement between the bootstrap and $\epsilon$-expansion results. The scaling dimensions corresponding to the minimum of the transformed navigator function follow the epsilon-expansion curves especially well, confirming the hypothesis of [1] that it provides a better estimate of the true scaling dimensions than a generic allowed point. Of course the error bars on the bootstrap results should be taken with a grain of salt, since there is no systematic way to account for the non-unitarity of the fractional-$d$ solution to crossing. We can only comment that, just like in the case of the Ising model in fractional dimensions [11], these non-unitary effects seem insignificant (at our level of precision), since the agreement with the $\epsilon$-expansion stays excellent for all values of $d$.

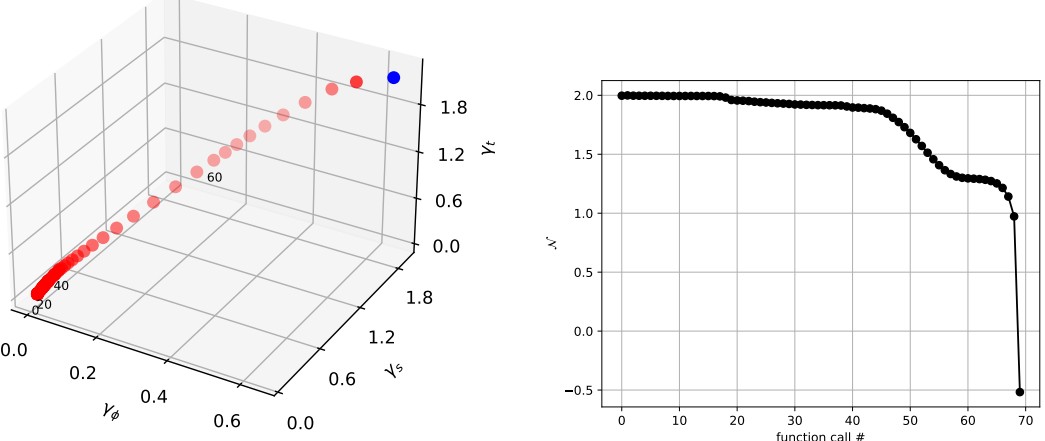

FIG. 6.   BFGS run for $N = 2$, $d = 3.7$ at $\Lambda = 19$ using as the initial guess the minimum of the run at $d = 3.8$. Red points are disallowed, blue points are allowed. From the monotonic increase of $\gamma_\phi, \gamma s$ and $\gamma_t$ to large values, we see that this run misses the (tiny) island and runs off to the peninsula.

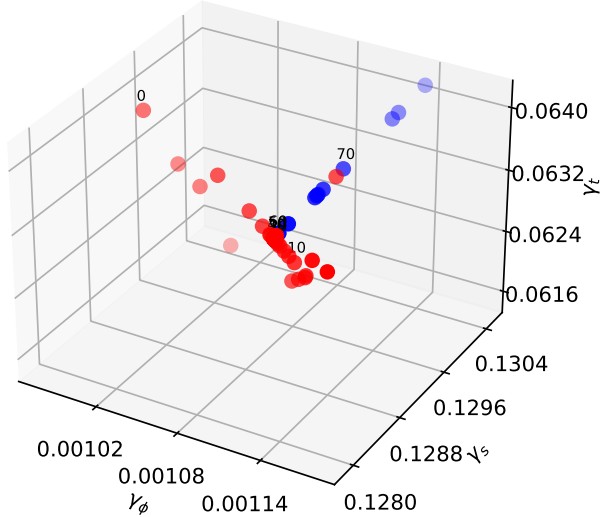

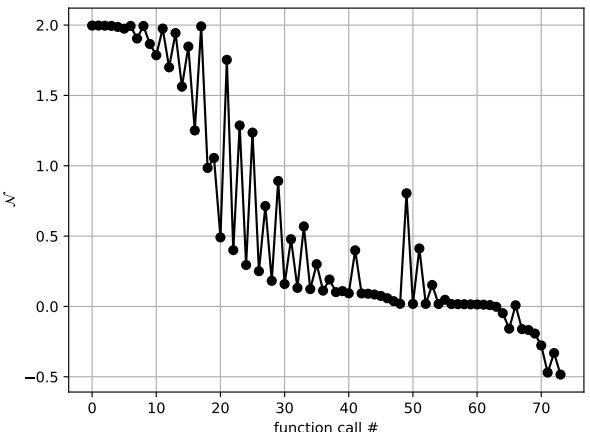

FIG. 7. BFGS run for $N = 2$, $d = 3.7$ at $\Lambda = 19$ with the initial guess supplied by Eq. (16). Red points are disallowed, blue points are allowed. This run does converge to the island.

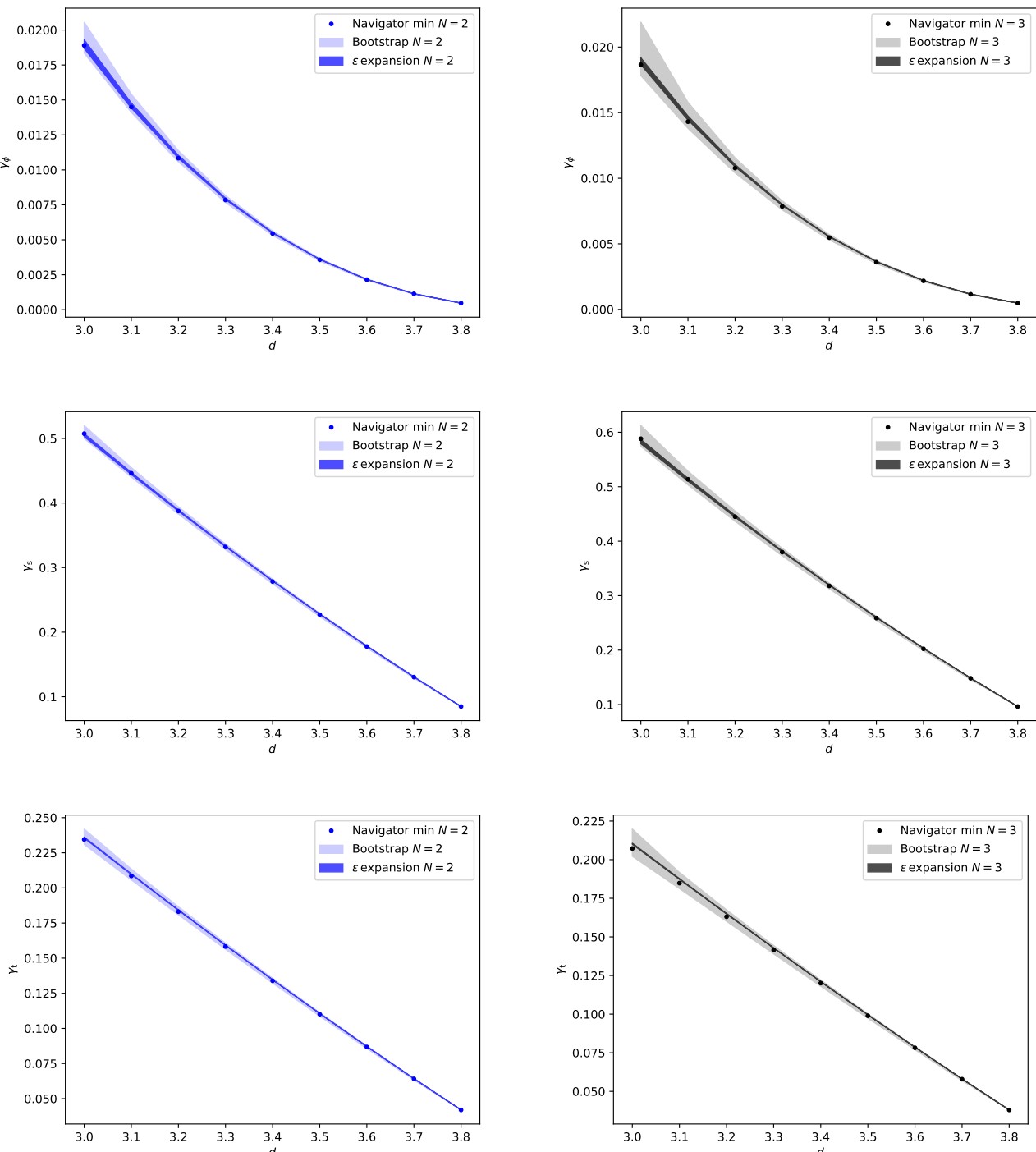

FIG. 8. $\gamma_\phi$, $\gamma_s$ and $\gamma_t$ as functions of $d$ for $N = 2, 3$, as determined by both the conformal bootstrap with the help of the navigator function, and from the resummation of 6-loop $\epsilon$-expansions.

| | Bootstrap | | | Resummed $4-\epsilon$ expansion | | |
|---|---|---|---|---|---|---|
| $d$ | $\gamma_\phi$ | $\gamma_s$ | $\gamma_t$ | $\gamma_\phi$ | $\gamma_s$ | $\gamma_t$ |
| | $N=2$ | | | | | |
| 3.8 | $0.000471^{+0.000003}_{-0.000010}$ | $0.0848^{+0.0003}_{-0.0010}$ | $0.04196^{+0.00013}_{-0.00047}$ | $0.00047209$ $\pm0.00000007$ | $0.084937$ $\pm0.000002$ | $0.0420064$ $\pm0.0000005$ |
| 3.7 | $0.001135^{+0.000015}_{-0.000028}$ | $0.1304^{+0.0007}_{-0.0017}$ | $0.0641^{+0.0003}_{-0.0008}$ | $0.0011403$ $\pm0.0000007$ | $0.13067$ $\pm0.00002$ | $0.064231$ $\pm0.000003$ |
| 3.6 | $0.00215^{+0.00004}_{-0.00005}$ | $0.1777^{+0.0014}_{-0.0022}$ | $0.0868^{+0.0007}_{-0.0011}$ | $0.002165$ $\pm0.000003$ | $0.17839$ $\pm0.00006$ | $0.087120$ $\pm0.000009$ |
| 3.5 | $0.00357^{+0.00009}_{-0.00008}$ | $0.227^{+0.002}_{-0.003}$ | $0.1101^{+0.0012}_{-0.0013}$ | $0.003601$ $\pm0.000009$ | $0.22802$ $\pm0.00014$ | $0.11061$ $\pm0.00002$ |
| 3.4 | $0.00545^{+0.00018}_{-0.00013}$ | $0.278^{+0.003}_{-0.003}$ | $0.1339^{+0.0017}_{-0.0016}$ | $0.00550$ $\pm0.00002$ | $0.2796$ $\pm0.0003$ | $0.13465$ $\pm0.00005$ |
| 3.3 | $0.00785^{+0.00031}_{-0.00018}$ | $0.332^{+0.005}_{-0.004}$ | $0.158^{+0.002}_{-0.002}$ | $0.00793$ $\pm0.00005$ | $0.3330$ $\pm0.0006$ | $0.15921$ $\pm0.00009$ |
| 3.2 | $0.0108^{+0.0005}_{-0.0003}$ | $0.388^{+0.007}_{-0.005}$ | $0.183^{+0.003}_{-0.002}$ | $0.01094$ $\pm0.00010$ | $0.3884$ $\pm0.0010$ | $0.18427$ $\pm0.00014$ |
| 3.1 | $0.0145^{+0.0009}_{-0.0004}$ | $0.446^{+0.009}_{-0.006}$ | $0.209^{+0.005}_{-0.003}$ | $0.01460$ $\pm0.00018$ | $0.4457$ $\pm0.0015$ | $0.2098$ $\pm0.0002$ |
| 3 | $0.0189^{+0.0017}_{-0.0006}$ | $0.507^{+0.012}_{-0.007}$ | $0.234^{+0.007}_{-0.004}$ | $0.0190$ $\pm0.0003$ [18] | $0.505$ $\pm0.002$ [18] | $0.2358$ $\pm0.0003$ [17] |
| | $N=3$ | | | | | |
| 3.8 | $0.000482^{+0.000003}_{-0.000013}$ | $0.0964^{+0.0003}_{-0.0013}$ | $0.03798^{+0.00012}_{-0.00051}$ | $0.00048301$ $\pm0.00000006$ | $0.096536$ $\pm0.000004$ | $0.0380293$ $\pm0.0000004$ |
| 3.7 | $0.001155^{+0.000017}_{-0.000034}$ | $0.1482^{+0.0009}_{-0.0023}$ | $0.0579^{+0.0003}_{-0.0009}$ | $0.0011621$ $\pm0.0000006$ | $0.14865$ $\pm0.00002$ | $0.058046$ $\pm0.000002$ |
| 3.6 | $0.00218^{+0.00005}_{-0.00006}$ | $0.202^{+0.002}_{-0.003}$ | $0.0782^{+0.0008}_{-0.0012}$ | $0.002199$ $\pm0.000003$ | $0.20318$ $\pm0.00009$ | $0.078591$ $\pm0.000008$ |
| 3.5 | $0.00360^{+0.00011}_{-0.00011}$ | $0.259^{+0.003}_{-0.004}$ | $0.0990^{+0.0013}_{-0.0016}$ | $0.003644$ $\pm0.000008$ | $0.2601$ $\pm0.0002$ | $0.09960$ $\pm0.00002$ |
| 3.4 | $0.00547^{+0.00023}_{-0.00017}$ | $0.318^{+0.005}_{-0.005}$ | $0.120^{+0.002}_{-0.002}$ | $0.005549$ $\pm0.000019$ | $0.3195$ $\pm0.0005$ | $0.12103$ $\pm0.00005$ |
| 3.3 | $0.0079^{+0.0004}_{-0.0003}$ | $0.380^{+0.007}_{-0.007}$ | $0.141^{+0.003}_{-0.002}$ | $0.00797$ $\pm0.00004$ | $0.3814$ $\pm0.0009$ | $0.14284$ $\pm0.00009$ |
| 3.2 | $0.0108^{+0.0008}_{-0.0004}$ | $0.445^{+0.010}_{-0.008}$ | $0.163^{+0.004}_{-0.003}$ | $0.01096$ $\pm0.00008$ | $0.4459$ $\pm0.0016$ | $0.16500$ $\pm0.00015$ |
| 3.1 | $0.0143^{+0.0015}_{-0.0005}$ | $0.514^{+0.015}_{-0.010}$ | $0.185^{+0.007}_{-0.004}$ | $0.01459$ $\pm0.00015$ | $0.513$ $\pm0.003$ | $0.1875$ $\pm0.0002$ |
| 3 | $0.0187^{+0.0032}_{-0.0008}$ | $0.588^{+0.024}_{-0.014}$ | $0.207^{+0.013}_{-0.005}$ | $0.0189$ $\pm0.0003$ [18] | $0.583$ $\pm0.004$ [18] | $0.2103$ $\pm0.0003$ |

TABLE I. Comparison of bootstrap and $\epsilon$-expansion results for the anomalous dimensions of the first scalar $O(N)$ vector, scalar singlet and scalar traceless-symmetric 2-index tensor, for $N = 2, 3$ and $d \in [3, 4]$. Bootstrap results are given as the values at the transformed navigator minimum, with uncertainties given by the distance to the maximal and minimal values determined by the Constrained BFGS algorithm. Cited values are taken from Fig. 2 of [17] or deduced from the ancillary file "resummation.pdf" of [18].

A crucial step in following a bootstrap solution is to verify that there are no low-lying operators that dangerously approach the gaps we impose (if there were, modifications to our gap assumptions would have to be made). We show in Figs. 9 and 10 the dimensions of the subleading operators $\phi'$, $t'$, $J'_\mu$ and $T'_{\mu\nu}$ as determined by the Extremal Functional Method (EFM) [46] at each navigator minimum [4]. Most of the operators stay well above the gap assumptions made, as expected from the $\epsilon$-expansion (here we show the unresummed expansions given in the recent review article [25] [5]). Only $t'$ near $d = 4$ has a dimension close to the gap assumed, but from the top right of Fig. 9 it is clear that the gap assumption is never violated.

Something strange happens in the spin-0 $S$ sector. We observe that when the spectrum is extracted at the navigator minimum, the expected operator $s'$ appears lower than predicted by the $\epsilon$-expansion for most points, with $s'$ and $s''$ then respectively just below and just above the prediction of the $\epsilon$-expansion. We show this behaviour for the $N = 2$ case in Fig. 11. As presented in the bottom plot of Fig. 11, this behaviour happens when we are close enough to the GFF minimum for the test point $(d, N) = (3.4, 2)$, with allowed points further away from the GFF minimum being in much better agreement with the $\epsilon$-expansion. It would be interesting to see if this effect disappears by increasing the derivative order $\Lambda$ enough, and if this is not the case, to understand if and how this splitting is related to the GFF solution, but a high precision analysis of this question is beyond the scope of this work. Notwithstanding this strange behaviour in $s'$, we've firmly established that the gap assumptions made in this section were always consistent with the actual spectrum of the $O(2)$ and $O(3)$ models.

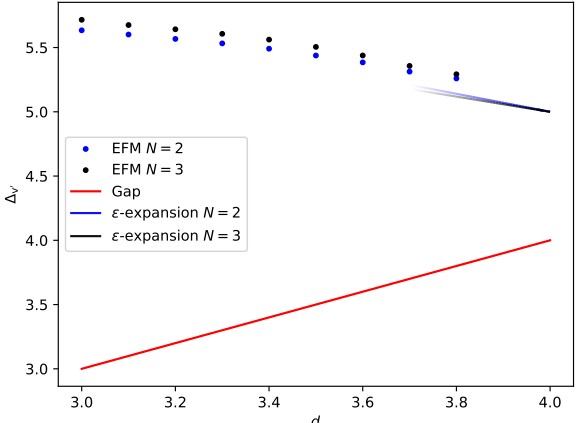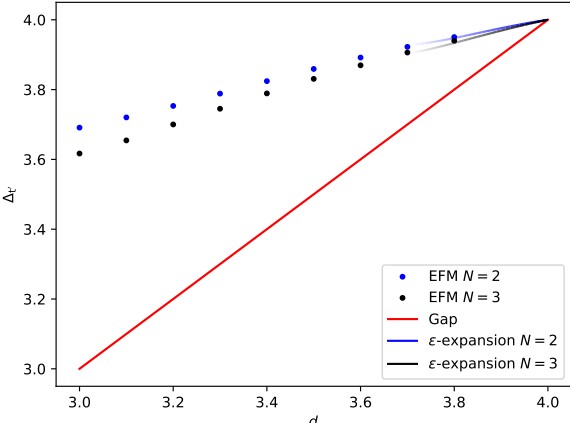

FIG. 9.    Subleading operators in 4 of the 5 channels where gap assumptions were made (continues in Fig. 10). Left: Spin-0 $V$ channel. The larger discrepancy with the $\epsilon$-expansion (the operators even appear in the opposite order as they do in the $\epsilon$-expansion) is not surprising for operators with large twist in a setup with relatively low $\Lambda$. Right: Spin-0 $T$ channel.

---

[4] Throughout this paper, unless otherwise stated, we maximize the stress-tensor OPE (after eliminating one of $(\lambda_{\phi\phi T_{\mu\nu}}, \lambda_{ss T_{\mu\nu}})$ using Ward identities) in order to extract the spectrum. Even though Ward identities were not imposed in (10), the navigator minima were deep enough in the allowed region that they were still allowed after imposing Ward identities.

[5] $T'_{\mu\nu}$ is compared to the second subleading operator of [25] (in their notation, Op[S,2,3]); the EFM in this channel missed the true subleading operator Op[S,2,2] at both $N = 2$ and $N = 3$, most likely because it is close in dimension to the second subleading operator (both are of the form $\partial_\mu \partial_\nu \phi_S^4$ in the $\epsilon$-expansion) and its OPE coefficients are small in comparison. From the ancillary file of [25], we have for example the ratios of their two OPE coefficients at $(d = 4, N = 2)$ given by $\frac{\lambda_{\phi\phi\text{Op[S,2,2]}}}{\lambda_{\phi\phi\text{Op[S,2,3]}}} \xrightarrow{d\to 4} 0.12$ and $\frac{\lambda_{ss\text{Op[S,2,2]}}}{\lambda_{ss\text{Op[S,2,3]}}} \approx 0.41$. We thank J. Henriksson for pointing this out.

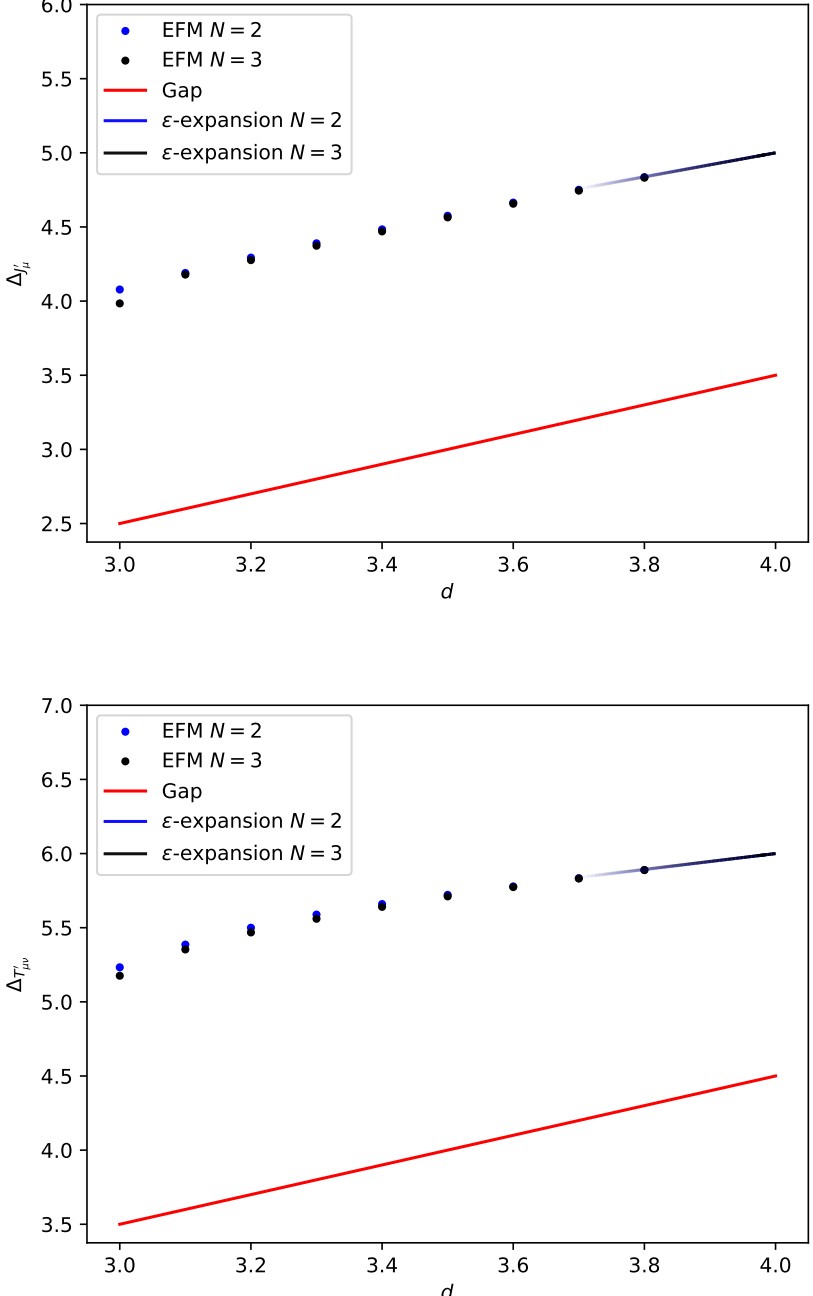

FIG. 10. Subleading operators in 4 of the 5 channels where gap assumptions were made (continued from Fig. 9). Top: Spin-1 $A$ channel. Bottom: Spin-2 $S$ channel.

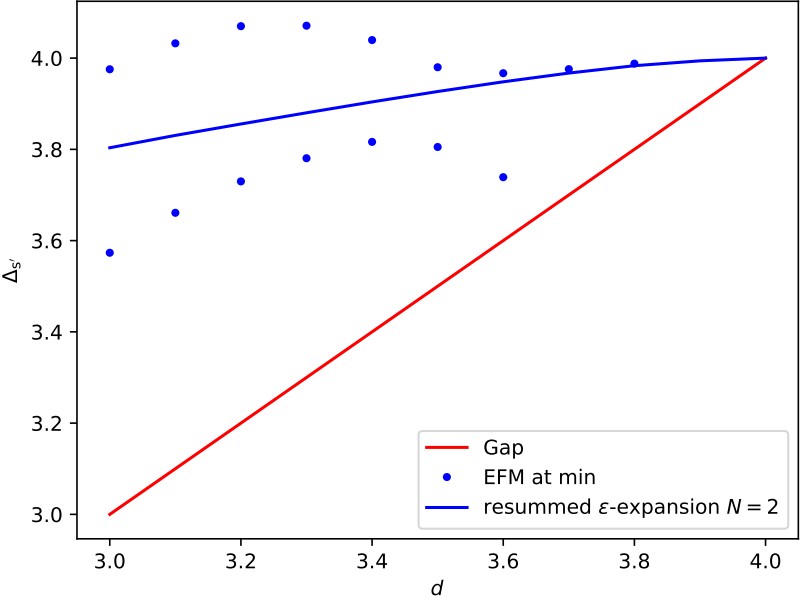

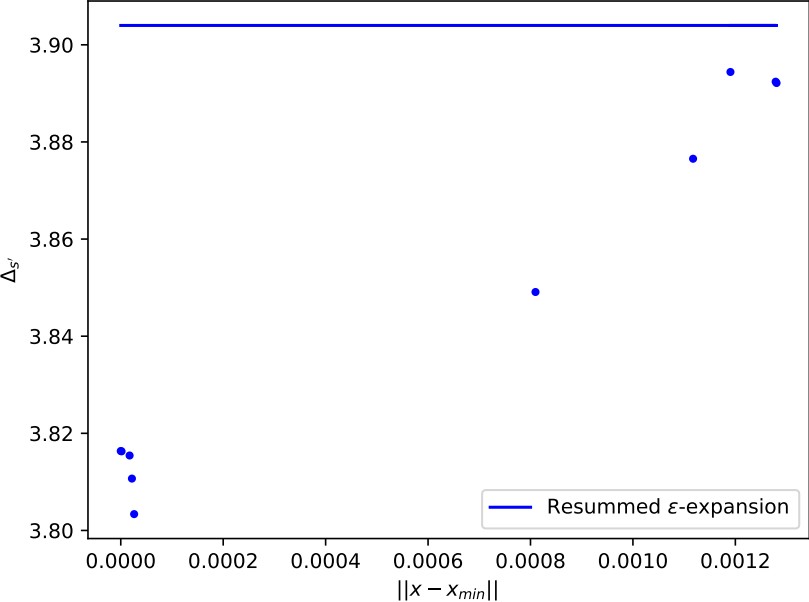

FIG. 11. Top: Jump in subleading operator $s'$ in the $\ell = 0$ $S$ channel found by the EFM at the navigator minimum for $N = 2$. Bottom: Evolution of $s'$ extracted from different points along the BFGS run for $(d = 3.4, N = 2)$, as functions of their distance to the transformed navigator minimum.

## VI.  SAILING THROUGH N

Non-unitarity due to fractional values of $d$ did not influence the low-lying part of the spectrum of the $O(N)$ models determined by the unitary bootstrap. This was expected, as [2] showed for the case of scalar $\phi^4$ theory that unitarity-violating operators appear in fractional dimensions only at large scaling dimensions. The fractional-$N$ $O(N)$ models are also expected to be non-unitary, with the operators at the source of the non-unitarity having generically large scaling dimensions for large values of $N$ [3]. Still, a previous single-correlator analysis indicated an impressive match of the unitary bootstrap with RG and Monte Carlo methods right down to the limit $N \to 0$ [47]. It is natural to wonder if we might see signs of non-unitarity for fractional values of $N$ when extending to a mixed-correlator setup. Therefore, using again the setup (10) with gaps $\vec{\Delta}^* = (3, 3, 3, 2.5, 3.5)$, we repeat an analysis similar to Section V, this time in $d = 3$ and exploring the range $N \in [1, 3]$. The results are presented in Table II and Fig. 12. We see that the bootstrap results are always consistent with the $\epsilon$-expansion, but have larger error bars because of the relatively low derivative order $\Lambda = 19$. Although the bootstrap uncertainties are bigger here than they were in Section V, the agreement is just as good in the limit $N \to 1$, where the $O(N)$ islands decrease significantly in size. The location of the transformed navigator minimum, just like it was the case in Section V, follows the $\epsilon$-expansion quite closely for all three scaling dimensions. We are therefore confident in asserting that the bootstrap at this derivative order is insensitive to non-unitarity in this range of $N$. We will see shortly that contrary to the results above, non-unitarity will drastically influence the fate of the $O(N)$ islands below this range.

We would like to make one comment before proceeding: we have presented a way to substantially cut down on the cost of following a bootstrap solution through an external parameter space with the navigator by using Eq. (16). The simple trick of using Eq. (16) is just one example of how one may use the additional information (as compared to the binary information provided by the usual scanning-based methods) encoded in the continuous navigator function to speed-up conformal bootstrap calculations. To demonstrate the efficacy of this method and to obtain the results we were after in this paper, it was enough for us to consider the crossing equation at a relatively low derivative order $\Lambda = 19$. But of course, one could have also been interested in getting to the navigator minimum at a much higher $\Lambda$ if he/she wanted to obtain very precise estimates of a certain amount of CFT data, and he/she would have benefited from Eq. (16) just the same. Furthermore, in that case, using a trick presented in Section 5.4 of [1], he/she could decrease his/her computational time by minimizing the navigator at a relatively low $\Lambda$ first, and then using the navigator minimum and final Hessian at this low $\Lambda$ as the starting point for a higher $\Lambda$ calculation. By iteratively going up in $\Lambda$ like this, he/she would cut down on the number of more expensive function calls at the higher $\Lambda$'s. For example, going to the $\Lambda = 27$ minimum at $(d, N) = (3, 2)$ using the $\Lambda = 19$ minimum and final Hessian as the initial input only required 13 function calls, confirming the efficiency of the trick. Of course, in using the navigator method as a substitute for the usual scanning-based methods, he/she would have to deal with the fact that each navigator evaluation is more expensive than running SDPB on feasibility mode [6], with the very large-$\Lambda$ navigator evaluations possibly becoming quite expensive.

---

[6] For example, for one point tested in the $\Lambda = 19$ $O(N = 1)$ island, running SDPB on feasibility mode with the option detectPrimalFeasibleJump was about 3.8 times faster than computing the navigator with dualityGapThreshold $= 10^{-25}$ at that point for an otherwise identical setup.

| | Bootstrap | | | Resummed $4 - \epsilon$ expansion | | |
|---|---|---|---|---|---|---|
| $N$ | $\Delta_\phi$ | $\Delta_s$ | $\Delta_t$ | $\Delta_\phi$ | $\Delta_s$ | $\Delta_t$ |
| 3 | $0.5187\,^{+0.0032}_{-0.0008}$ | $1.588\,^{+0.024}_{-0.014}$ | $1.207\,^{+0.013}_{-0.005}$ | $0.5189 \pm 0.0003$ [18] | $1.583 \pm 0.004$ [18] | $1.2103 \pm 0.0003$ |
| 2.9 | $0.5187\,^{+0.0030}_{-0.0008}$ | $1.581\,^{+0.023}_{-0.013}$ | $1.210\,^{+0.012}_{-0.005}$ | $0.5189 \pm 0.0003$ | $1.576 \pm 0.004$ | $1.2127 \pm 0.0003$ |
| 2.8 | $0.5188\,^{+0.0029}_{-0.0008}$ | $1.573\,^{+0.021}_{-0.013}$ | $1.212\,^{+0.011}_{-0.005}$ | $0.5190 \pm 0.0003$ | $1.569 \pm 0.004$ | $1.2151 \pm 0.0002$ |
| 2.7 | $0.5188\,^{+0.0027}_{-0.0008}$ | $1.565\,^{+0.020}_{-0.012}$ | $1.215\,^{+0.011}_{-0.005}$ | $0.5190 \pm 0.0003$ | $1.561 \pm 0.004$ | $1.2176 \pm 0.0003$ |
| 2.6 | $0.5188\,^{+0.0025}_{-0.0007}$ | $1.557\,^{+0.019}_{-0.011}$ | $1.218\,^{+0.010}_{-0.005}$ | $0.5191 \pm 0.0003$ | $1.553 \pm 0.003$ | $1.2201 \pm 0.0003$ |
| 2.5 | $0.5189\,^{+0.0024}_{-0.0007}$ | $1.549\,^{+0.018}_{-0.011}$ | $1.220\,^{+0.010}_{-0.004}$ | $0.5191 \pm 0.0003$ | $1.546 \pm 0.003$ | $1.2226 \pm 0.0003$ |
| 2.4 | $0.5189\,^{+0.0022}_{-0.0007}$ | $1.541\,^{+0.017}_{-0.010}$ | $1.223\,^{+0.009}_{-0.004}$ | $0.5191 \pm 0.0003$ | $1.538 \pm 0.003$ | $1.2252 \pm 0.0003$ |
| 2.3 | $0.5189\,^{+0.0021}_{-0.0006}$ | $1.533\,^{+0.016}_{-0.009}$ | $1.226\,^{+0.009}_{-0.004}$ | $0.5191 \pm 0.0003$ | $1.530 \pm 0.003$ | $1.2278 \pm 0.0003$ |
| 2.2 | $0.5189\,^{+0.0019}_{-0.0006}$ | $1.525\,^{+0.015}_{-0.009}$ | $1.229\,^{+0.008}_{-0.004}$ | $0.5190 \pm 0.0003$ | $1.522 \pm 0.003$ | $1.2304 \pm 0.0004$ |
| 2.1 | $0.5189\,^{+0.0018}_{-0.0006}$ | $1.516\,^{+0.013}_{-0.008}$ | $1.232\,^{+0.008}_{-0.004}$ | $0.5190 \pm 0.0003$ | $1.513 \pm 0.003$ | $1.2331 \pm 0.0003$ |
| 2 | $0.5189\,^{+0.0017}_{-0.0006}$ | $1.507\,^{+0.012}_{-0.007}$ | $1.234\,^{+0.007}_{-0.004}$ | $0.5190 \pm 0.0003$ [18] | $1.505 \pm 0.002$ [18] | $1.2358 \pm 0.0003$ [17] |
| 1.9 | $0.5189\,^{+0.0015}_{-0.0005}$ | $1.499\,^{+0.011}_{-0.007}$ | $1.237\,^{+0.007}_{-0.003}$ | $0.5190 \pm 0.0003$ | $1.4965 \pm 0.0024$ | $1.2385 \pm 0.0004$ |
| 1.8 | $0.5188\,^{+0.0014}_{-0.0005}$ | $1.490\,^{+0.010}_{-0.006}$ | $1.240\,^{+0.006}_{-0.003}$ | $0.5189 \pm 0.0003$ | $1.487 \pm 0.002$ | $1.2413 \pm 0.0004$ |
| 1.7 | $0.5188\,^{+0.0013}_{-0.0005}$ | $1.480\,^{+0.009}_{-0.006}$ | $1.243\,^{+0.006}_{-0.003}$ | $0.5188 \pm 0.0003$ | $1.479 \pm 0.002$ | $1.2441 \pm 0.0005$ |
| 1.6 | $0.5188\,^{+0.0011}_{-0.0004}$ | $1.471\,^{+0.008}_{-0.005}$ | $1.246\,^{+0.005}_{-0.003}$ | $0.5188 \pm 0.0003$ | $1.470 \pm 0.002$ | $1.2469 \pm 0.0003$ |
| 1.5 | $0.5187\,^{+0.0010}_{-0.0004}$ | $1.462\,^{+0.007}_{-0.005}$ | $1.250\,^{+0.005}_{-0.003}$ | $0.5187 \pm 0.0003$ | $1.460 \pm 0.002$ | $1.2499 \pm 0.0003$ |
| 1.4 | $0.5186\,^{+0.0009}_{-0.0003}$ | $1.452\,^{+0.006}_{-0.004}$ | $1.253\,^{+0.004}_{-0.003}$ | $0.5186 \pm 0.0003$ | $1.451 \pm 0.002$ | $1.2527 \pm 0.0006$ |
| 1.3 | $0.5185\,^{+0.0007}_{-0.0003}$ | $1.443\,^{+0.005}_{-0.004}$ | $1.256\,^{+0.004}_{-0.002}$ | $0.5185 \pm 0.0003$ | $1.4408 \pm 0.0015$ | $1.2557 \pm 0.0006$ |
| 1.2 | $0.5184\,^{+0.0006}_{-0.0003}$ | $1.433\,^{+0.004}_{-0.003}$ | $1.259\,^{+0.003}_{-0.002}$ | $0.5184 \pm 0.0003$ | $1.4311 \pm 0.0014$ | $1.2587 \pm 0.0006$ |
| 1.1 | $0.5183\,^{+0.0004}_{-0.0002}$ | $1.423\,^{+0.003}_{-0.003}$ | $1.263\,^{+0.002}_{-0.002}$ | $0.5182 \pm 0.0003$ | $1.4209 \pm 0.0012$ | $1.2617 \pm 0.0005$ |
| 1 | $0.518180^*$ | $1.41296^*$ | $1.26595^*$ | $0.5181 \pm 0.0003$ [18] | $1.4108 \pm 0.0011$ [18] | $1.2648 \pm 0.0006$ [17] |

TABLE II. Same as Table I, for the scaling dimensions instead of the anomalous dimensions, for $d = 3$ and $N \in [1, 3]$. Bootstrap results are given as the values at the transformed navigator minimum, with uncertainties given by the distance to the maximal and minimal values determined by the Constrained BFGS algorithm (the values at the navigator minimum are given with asterisks in cases where uncertainties were not calculated). Cited values are taken from Fig. 2 of [17] or deduced from the ancillary file "resummation.pdf" of [18].

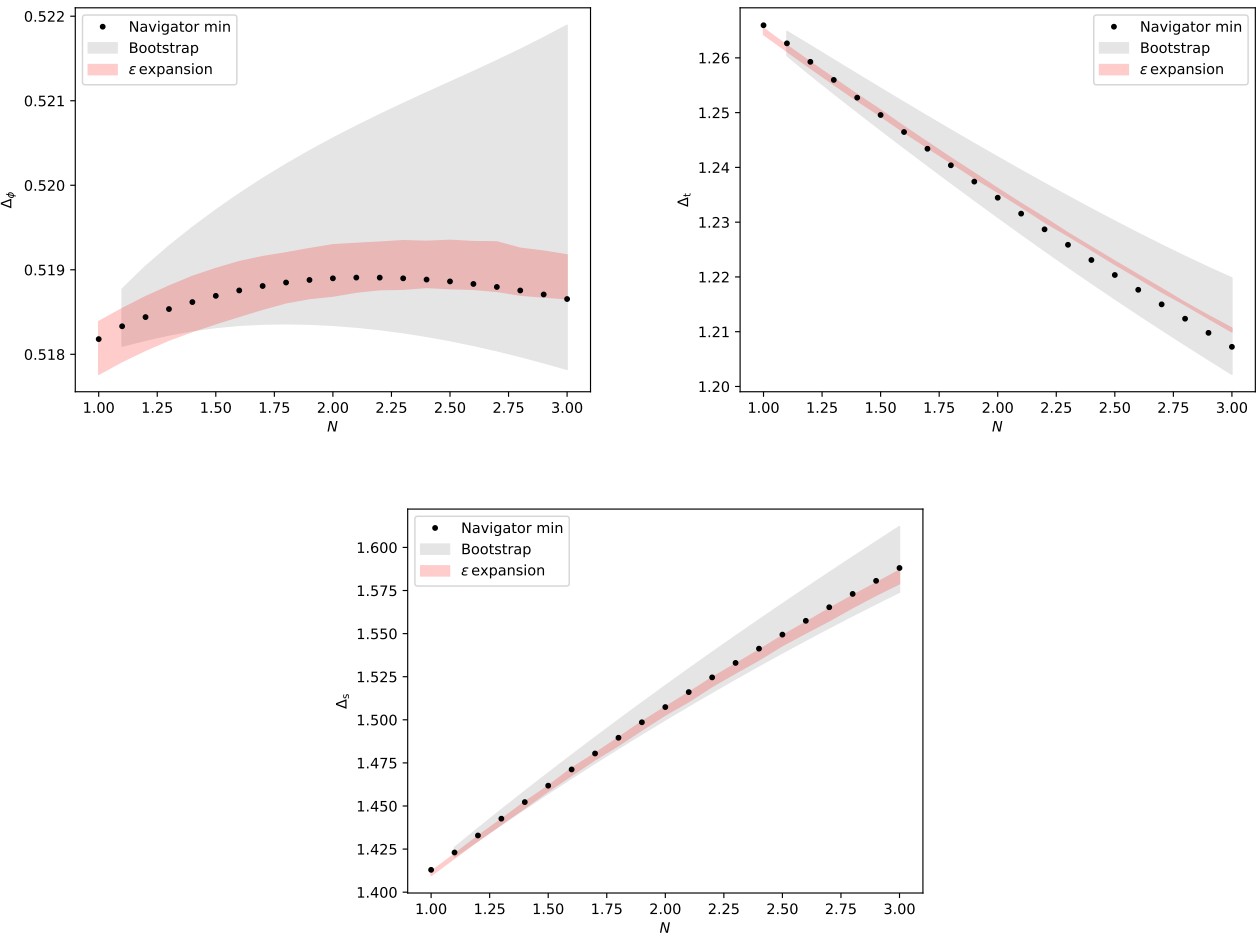

FIG. 12. $\Delta_\phi$, $\Delta_s$ and $\Delta_t$ as functions of $N$, as determined by both the conformal bootstrap with the help of the navigator function, and from the resummation of 6-loop $\epsilon$-expansions.

## A. $\mathbf{N \to 1}$ limit and sinking of the O(N) islands

Table II indicates that the mixed-correlator bootstrap correctly captures the limit $N \to 1$, and enables one to determine the dimensions of operators like $\Delta_t$ which would be invisible in the bootstrap of the Ising Model. What happens at exactly $N = 1$ is worth further consideration. As already noted in [47], the full system of crossing equations at $N = 1$ contains the Ising crossing equations, where the $O(N = 1)$ $V$ and $S$ sectors correspond respectively to the Ising $\mathbb{Z}_2$-odd and $\mathbb{Z}_2$-even sectors. Indeed, one concludes from the crossing vectors (5) that the lines $\{1+2, 4, 5, 6, 7\}$ in the $V$ and $S$ crossing vectors for $N = 1$ are simply those of the Ising mixed-correlator $\mathbb{Z}_2$-odd and $\mathbb{Z}_2$-even vectors, as given e.g. in (3.12) of [5]. This combination of lines renders the contribution from the $A$ and $T$ sectors identically zero, completing the identification to the Ising crossing equations. The fact that the crossing vectors separate into "Ising + rest" implies a similar

separation of the navigator problem. Indeed, our $O(N)$ navigator problem

$$\min_{(\Delta_\phi,\Delta_s,\Delta_t)} \lambda \quad | \quad \sum_{\mathcal{R}\in\{V,S,A,T\}} \sum_{\mathcal{O}\in\mathcal{R}} \vec{\lambda}^\intercal_{\mathcal{O}_\mathcal{R}} \cdot \vec{V}_{\mathcal{R},\Delta_\mathcal{O},\ell_\mathcal{O}} \cdot \vec{\lambda}_{\mathcal{O}_\mathcal{R}} = -\lambda\vec{M}_{\mathrm{GFF}} \tag{18}$$

becomes at $N = 1$

$$\min_{(\Delta_\phi,\Delta_s,\Delta_t)} \lambda \quad \text{such that}$$
$$\begin{cases} \sum_{\mathcal{R}\in\{V,S\}} \sum_{\mathcal{O}\in\mathcal{R}} \vec{\lambda}^\intercal_{\mathcal{O}_\mathrm{R}} \cdot \vec{V}^{\mathbb{Z}_2}_{\mathcal{R},\Delta_\mathcal{O},\ell_\mathcal{O}} \cdot \vec{\lambda}_{\mathcal{O}_\mathcal{R}} = -\lambda\vec{M}^{\mathbb{Z}_2}_{\mathrm{GFF}} \\ \sum_{\mathcal{R}\in\{V,S,A,T\}} \sum_{\mathcal{O}\in\mathcal{R}} \vec{\lambda}^\intercal_{\mathcal{O}_\mathrm{R}} \cdot \vec{\tilde{V}}_{\mathcal{R},\Delta_\mathcal{O},\ell_\mathcal{O}} \cdot \vec{\lambda}_{\mathcal{O}_\mathcal{R}} = -\lambda\vec{\tilde{M}}_{\mathrm{GFF}} \end{cases} . \tag{19}$$

Crossing vectors with a $\mathbb{Z}_2$ superscript are 5-component vectors made up of lines $\{1+2,4,5,6,7\}$ of the full $O(N=1)$ vectors. Crossing vectors with a tilde are two-component vectors made up of the rest of the full $O(N=1)$ vectors (so for example, lines 1 and 3). The first condition states that the $V$ and $S$ sectors should solve the Ising crossing equation augmented by a GFF contribution $\lambda\vec{M}^{\mathbb{Z}_2}_{\mathrm{GFF}}$ containing all $V$ and $S$ GFF operators below the gap assumptions of (10). If the first condition was the only condition, the navigator would be independent of $\Delta_t$, and equal to the Ising navigator of [1] $\mathcal{N}^{\mathbb{Z}_2}(\Delta_\sigma,\Delta_\epsilon)$ under the identifications $(\Delta_\phi \leftrightarrow \Delta_\sigma, \Delta_s \leftrightarrow \Delta_\epsilon)$ if identical assumptions were made on the equivalent sectors. Because the tilde vectors are non-zero in the $A$ and $T$ sectors, we expect that the second condition of (19) will cause the navigator to depend on the assumptions made in these sectors. In particular, we will have some dependence of the navigator on $\Delta_t$ if $\Delta_t$ is to be constrained to a small finite region of allowed values at $N = 1$, as suggested by the limit $N \to 1$ in Fig. 12.

What we observe numerically is that for a given $(\Delta_\phi,\Delta_s)$, there is some range of $\Delta_t$ where the navigator is flat. As expected, for that range, (19) effectively reduces only to minimization over the Ising condition. Indeed, in minimizing the transformed navigator at $N = 1$, the navigator function was computed at 45 points, some determined to be allowed and some disallowed, and actually the gradients of the navigator function for all those points were found to be zero (to our numerical precision) in the $\Delta_t$ direction[7]. We have checked for a number of those points that the values of the navigator function match exactly those obtained in the pure Ising navigator setup of [1] when equivalent assumptions were made between the $V$ and $\mathbb{Z}_2$-odd and the $S$ and $\mathbb{Z}_2$-even sectors. One should however not conclude that the navigator function never depends on $\Delta_t$. For example, its dependence on $\Delta_t$ for a certain $(\Delta_\phi,\Delta_s)$ is shown in Fig. 13. There is some small range where the navigator is constant at its Ising value, and then it increases once it leaves this region. This picture makes sense: for any given $(\Delta_\phi,\Delta_s,\Delta_t)$, we have

$$\mathcal{N}(\Delta_\phi,\Delta_s,\Delta_t) = \lambda^{O(N=1)}_{\min} \geq \lambda^{\mathbb{Z}_2}_{\min} = \mathcal{N}^{\mathbb{Z}_2}(\Delta_\phi,\Delta_s) \quad , \tag{20}$$

since from (19), every solution of the $O(N=1)$ augmented crossing equations gives a $V$ and $S$ sector that solves the Ising augmented crossing equations. Furthermore, the equality in Eq. (20) is reached if and only if there is a solution of the second condition of (19) with $\lambda = \lambda^{\mathbb{Z}_2}_{\min}$, where the $V$ and $S$ sectors solve the first condition also with $\lambda = \lambda^{\mathbb{Z}_2}_{\min}$. In other words, the $O(N=1)$ navigator is equal to the Ising navigator if and only the second condition of (19) can be solved with $\lambda = \lambda^{\mathbb{Z}_2}_{\min}$ for a spectrum with $V$ and $S$ sectors that solve the pure Ising navigator problem.

---

[7] For example, $\nabla\mathcal{N}(0.518127\ldots,1.41243\ldots,1.26559\ldots) = (-16685.8, 1311.79, 1.14678\times10^{-39})$.

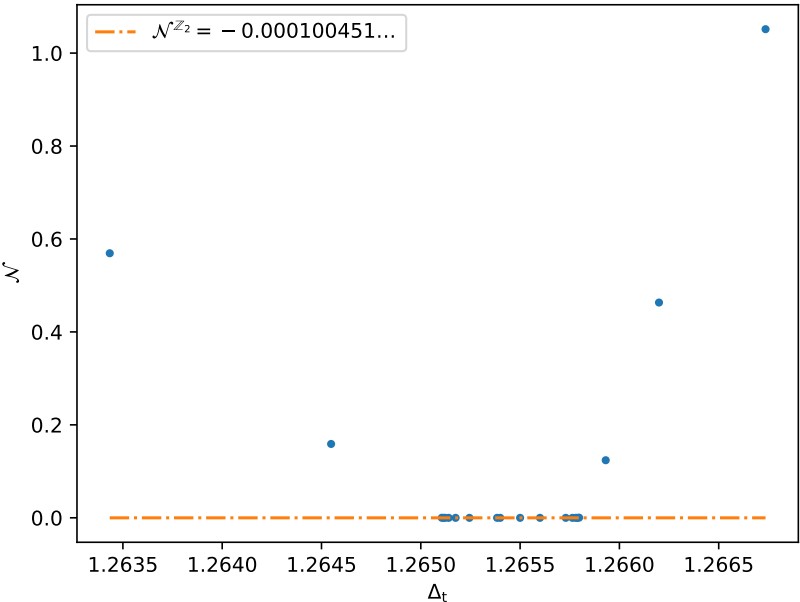

FIG. 13. Dependence of the navigator function on $\Delta_t$ at $N = 1$ for $(\Delta_\phi, \Delta_s) = (0.518139\ldots, 1.41252\ldots)$. The navigator function is constant in a small region, and equal in that region to the navigator function for the case of a mixed-correlator Ising setup where identical assumptions were made on the equivalent sectors $(V \leftrightarrow \mathbb{Z}_2-\text{odd} , S \leftrightarrow \mathbb{Z}_2-\text{even})$.

From the discussion above, we expect that the navigator function has a line of minima parametrized by $\Delta_t$. BFGS did in the end converge to one of those minima:

$$x^\star_{\min} = (0.518180\ldots, 1.41296\ldots, 1.26595\ldots) \quad . \tag{21}$$

We did not compute the full size of the allowed region. Nevertheless, we have from Eq. (20) that the allowed values of $(\Delta_\phi, \Delta_s)$ are contained in the $\Lambda = 19$ Ising island. From the observed narrow range displayed in Fig. 13 where the navigator function is constant in $\Delta_t$ for a given $(\Delta_\phi, \Delta_s)$, we also expect the allowed region to be small in the $\Delta_t$ direction. In light of this, it is no surprise that the location in $\Delta_t$ of the minimum found by BFGS matches quite well with $\Delta_t = 1.2648(6)$ predicted by the epsilon expansion [17].

There is one thing in the navigator computation at $N = 1$ that is worrying: the actual minimal value of the transformed navigator $f(x)$. We plot in Fig. 14 the value of $f(x)$ at its minimum near $N = 1$. Remember that $f(x)$ is positive iff the actual navigator function $\mathcal{N}(x)$ is positive. At $N = 1.5$, $f(x_{\min}) = -0.072910$, and $f(x)$ increases as $N \to 1^+$, reaching $-0.000064$ at $N = 1$. Thus the "most allowed" point according to the navigator construction gets dangerously close to becoming disallowed as $N \to 1^+$. This offers a clear sign that the $O(N)$ islands may disappear somewhere just below $N = 1$. Although we did not precisely determine the location of the disappearance, we can say that the islands seem to disappear for good when going well below $N = 1$. Indeed, for two values we tested well below $N = 1$, BFGS was not able to find an allowed point. For $N = 0.9$ and $N = 0.5$, BFGS found minima that both lie above $\mathcal{N}(x) = 0$

(see Fig. 14). We hypothesize that the islands do disappear at some critical $N_c$ close to $N = 1$, and that $N_c \xrightarrow{\Lambda \to \infty} 1$. We will see in the rest of this section that the $O(N)$ model must actually become very non-unitary below $N = 1$, which explains the observed disappearance of the islands.

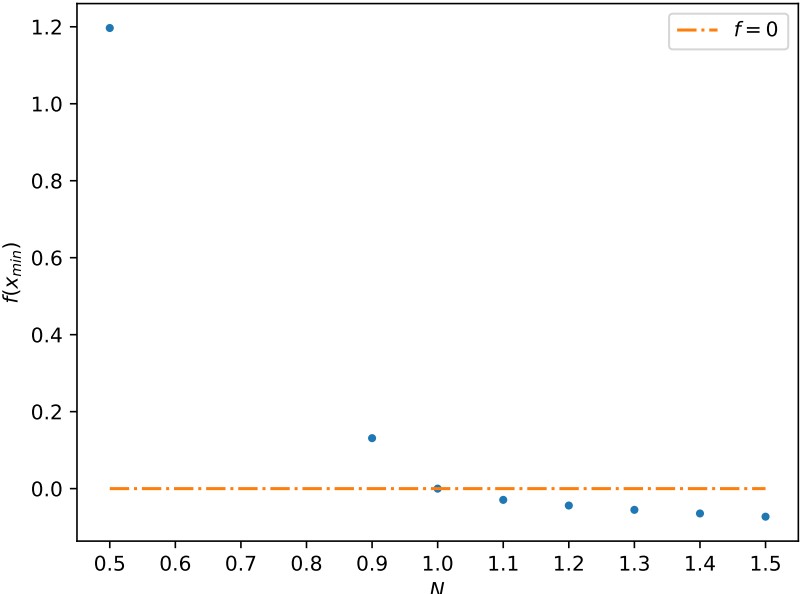

FIG. 14. Value of the transformed navigator (11) at its minimum, for $N$ between 0.5 and 1.5.

So why is it that the islands seem to disappear roughly at $N = 1$? Some non-unitarity was presumably already present in the high-lying part of the spectrum of the $O(N)$ model at fractional values of $N$ above $N = 1$ [3], but the bootstrap showed to be insensitive to this. Say we determine a unitary spectrum inside of the $\Lambda = 19$ $O(N)$ islands with the Extremal Functional Method as in the end of Section V, and follow this spectrum all the way down to $N = 1$. Why could we not just perturb the unitary spectrum we find at $N = 1$ to obtain a unitary one that solves the crossing equations just below $N = 1$? We present in Figs. 15 and 16 the fate of two operators which illustrate exactly why. Fig. 15 shows the $\phi \times s$ OPE coefficient of the lowest-lying $\ell = 1$ $V$ primary $v_\mu$ in the limit $N \to 1^+$. This OPE coefficient clearly goes to zero as $N$ tends to 1. Furthermore, from a fit of all values below $N = 1.2$, we find that the OPE coefficient goes to zero roughly as a square root: $\lambda_{\phi s v_\mu} \sim (N - 1)^{0.482834}$. We then show in Fig. 16 the fate of the subleading $\ell = 2$ $V$ primary $v'_{\mu\nu}$. We see in the right part of Fig. 16 that the lowest-lying $\ell = 2$ $V$ primary $v_{\mu\nu}$ continues at $N = 1$ into the lowest-lying $\ell = 2$ $\mathbb{Z}_2$-odd primary $\sigma_{\mu\nu}$ of the Ising model, while the third lowest-lying $\ell = 2$ $V$ primary $v''_{\mu\nu}$ continues approximately into $\sigma'_{\mu\nu}$ (the offset is due here to the lower derivative order used in comparison to [48]). The dimension of $v'_{\mu\nu}$ tends as $N$ goes to 1 to a dimension that does not correspond to any primary in the Ising spectrum. As expected, when we extract the spectrum at an allowed point in the $O(N = 1)$ island, we find that this primary has disappeared[8]. We show in the left part of Fig. 16 that this disappearance can be seen in the $\phi \times s$ OPE coefficient of $v'_{\mu\nu}$, which tends to zero as $N \to 1^+$, again roughly as a square

---

[8] We have encountered some problems in extracting the full spectrum exactly at $N = 1$. If we do so by extremizing

root (a fit of all values below $N = 1.2$ gives $\lambda_{\phi s v'_{\mu\nu}} \sim (N - 1)^{0.452306}$, although the fit is much less stable than that for $v_\mu$). We have therefore presented above two $V$ sector primaries, with relatively small scaling dimensions, whose $\phi \times s$ OPE coefficients behave approximately as square roots near $N = 1$. A naive continuation to the range $N < 1$ would require their squared OPE coefficients to become negative, violating unitarity. Thus, the presence of such primaries seemingly prevents us from obtaining a unitary solution to crossing at $\Lambda = 19$ below $N = 1$ by perturbing the unitary solution to crossing we get at $N = 1$. One may wonder if the problematic primaries are related to problematic primaries in the free $O(N)$ model, or in the $O(N)$ model in $4 - \epsilon$ dimensions, where a large amount of CFT data is known analytically [25]. We are going to see shortly that the answer to this question is yes.

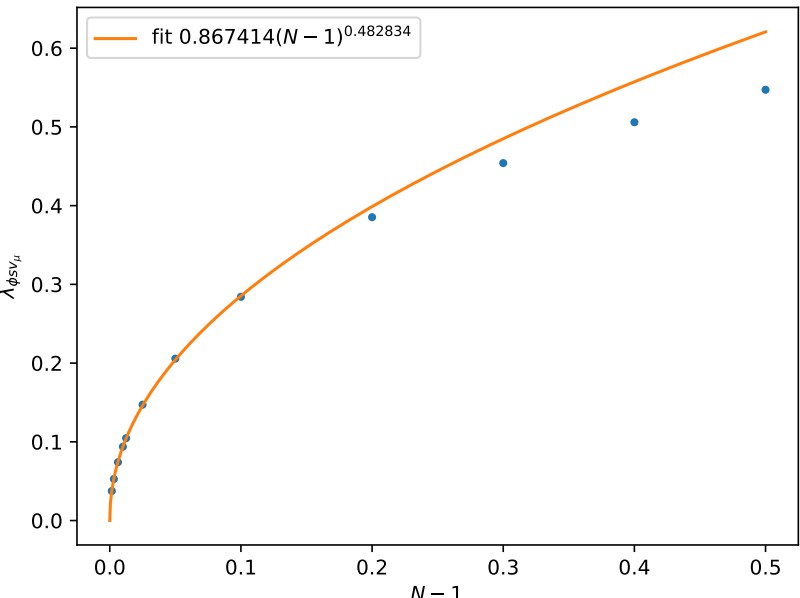

FIG. 15. Plot of the $\phi \times s$ OPE coefficient of the lowest-lying spin-1 $V$ primary, along with a fit of all values excluding those over $N = 1.1$.

some quantity in the channels present in the Ising crossing equation, SDPB looks for functionals $\vec{\alpha}$ such that $\vec{\alpha} \cdot \vec{V}_{T,\Delta,\ell} = \vec{\alpha} \cdot \vec{V}_{A,\Delta,\ell} = 0$ identically. Because of this, such an extremization cannot give any information on the $A$ and $T$ channels. When we instead try to extremize a quantity in one of these channels, e.g. by maximizing $\lambda_{\phi\phi t}$, the solution of the *primal* problem of SDPB (see [30] for its definition) has a large discontinuous jump when going from $N > 1$ to $N = 1$. Using this primal solution to determine OPE coefficients [48] results in none of the OPE coefficients we gather at $N = 1$ in the $A$ and $T$ sectors being sensible. We do not have a definitive explanation for this behaviour.

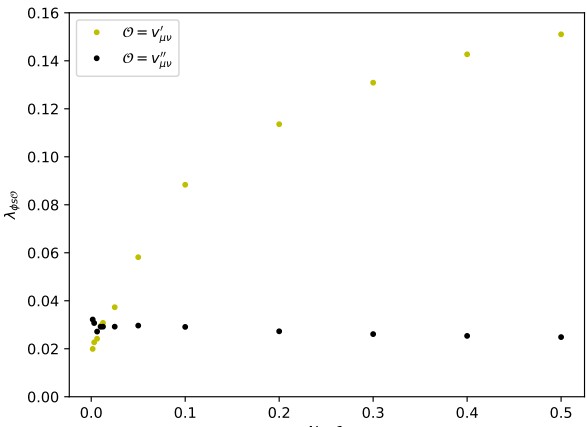 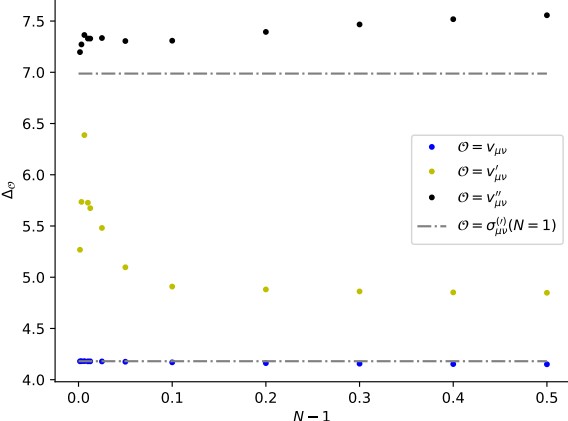

FIG. 16. Left: $\phi \times s$ OPE coefficient of the two subleading spin-2 $V$ primaries. Right: Scaling dimensions of the three lowest-lying spin-2 $V$ primaries. The straight lines indicate the dimensions of the two lowest-lying spin-2 $\mathbb{Z}_2$-odd primaries in the Ising Model according to [48].

Should we believe that the spectrum of the $O(N)$ model varies continuously from the free theory in $d = 4$ to the strongly-coupled theory in $d = 3$ (see [2] for a discussion), we might hope that the operators observed in $d = 3$ in Figs. 15 and 16 can be associated close to $d = 4$ to operators with similar behaviours in the limit $N \to 1$, and whose behaviours can be easily interpreted. From what will be presented below, it seems that this picture is valid. The lowest-lying $\ell = 1$ vector primary, which was the first operator discussed in the last paragraph, is given in the $\epsilon$-expansion by an operator with three fields and one derivative $v_\mu = \partial_\mu \phi_V^3$ according to Table 21 of [25]. At $N = 1$, this operator should become $\phi^2 \partial_\mu \phi$, which is a descendant of $\phi^3$. Therefore, the primary $v_\mu$ in $d = 3$ disappears in the limit $N \to 1^+$ just like $v_\mu$ in $d = 4 - \epsilon$ should. There are no accounts we could find in the literature of the OPE coefficient $\lambda_{\phi \phi^2 \partial_\mu \phi_V^3}(\epsilon)$, from which we could compare the approximate square root behavior observed in Fig. 15. However, another example of a primary operator becoming a descendant at $N = 1$ is the scalar singlet $\phi^2 (\partial \phi) \cdot (\partial \phi)$, which becomes a descendant of $\phi^4$ at $N = 1$ in the free theory as derived in [49]. In that case, (6.25) of [49] gives the free theory squared OPE coefficient

$$\lambda^2_{\phi^2 \, \phi^2 \, \phi^2 (\partial \phi) \cdot (\partial \phi)} = A \left( 4 - \frac{4}{N} \right) \qquad , \qquad A > 0 \quad . \tag{22}$$

This squared OPE coefficient crosses zero smoothly at $N = 1$ (and behaves like $(N - 1)$ around $N = 1$), meaning that the free $O(N)$ model gains some non-unitary below $N = 1$. Unfortunately, we could not really observe the disappearance of this operator in $d = 3$ with the bootstrap, most likely because it sits too high in the spectrum to be observed at $\Lambda = 19$.

The limit $N \to 1$ also sees many sets of different primary operators with the same classical dimensions merging into a lower number of primaries at $N = 1$. This is for example the case of the two lowest-lying $\ell = 2$ $V$ primaries $v_{\mu\nu}$ and $v'_{\mu\nu}$ (where $v'_{\mu\nu}$ was observed in Fig. 16 to disappear in $d = 3$). Table 21 of [25] gives these operators in the $\epsilon$-expansion as two operators of the form $\partial_\mu \partial_\nu \phi_V^3$. They become one single primary operator $\sigma_{\mu\nu} = \partial_\mu \partial_\nu \phi^3$ at $N = 1$ (see Table 14 of [25]). Another example discussed in [25] in the $\epsilon$-expansion is that of the two subleading $\ell = 2$ $S$

primaries, which are of the form $\partial_\mu \partial_\nu \phi_S^4$. They become a single primary $\partial_\mu \partial_\nu \phi^4$ at $N = 1$. In this case, the $\phi \times \phi$ squared OPE coefficients for both operators are known to $\mathcal{O}(\epsilon^2)$, and that of one of the two ([25] refer to it as $\mathcal{O}_{4,2,1}$) goes smoothly from positive to negative at $N = 1$. It is given in (3.15) of [25] as:

$$
\begin{aligned}
\lambda^2_{\phi\phi\mathcal{O}_{4,2,1}} &= \frac{N+2}{320N(N+8)^2} \frac{-76 + N + 3\sqrt{9N^2 - 8N + 624}}{\sqrt{9N^2 - 8N + 624}} \epsilon^2 + \mathcal{O}(\epsilon^3) \\
&= \left( \frac{1}{135000}(N-1) + \dots \right) \epsilon^2 + \mathcal{O}(\epsilon^3) \quad .
\end{aligned}
\tag{23}
$$

Many other primary operators disappearing at $N = 1$ because of this mechanism may be inferred from comparing Tables 8-14 to Tables 19-21 of [25]. In the cases we could find where data about OPE coefficients were known, disappearing primary operators of this sort had squared OPE coefficients that were analytic at $N = 1$, and that became negative below $N = 1$, making the $O(N)$ model in $d$ close to 4 very non-unitary. Fig. 16 shows that the problematic operators in $d \approx 4$ continue into problematic operators in $d = 3$.

We have shown above two different mechanisms (primaries becoming descendants and merging of primaries) which result in severe non-unitarity in the free $O(N)$ model and the $O(N)$ model in $d = 4 - \epsilon$ below $N = 1$, as many squared OPE coefficients involving low-lying primaries which were positive above $N = 1$ become negative below. We have seen in Figs. 15 and 16 that analogous behaviours could be observed in the spectra solving the $d = 3$ $O(N)$ crossing equations at $\Lambda = 19$ in the limit $N \to 1^+$. Therefore, to continue the solution to crossing below $N = 1$ would require many unitarity-violating contributions from low-lying operators, which explains the drastic difference between the results of the unitary bootstrap for $N > 1$ and $N < 1$. Finally, we would like to point out that the problematic operators we see in the limit $N \to 1$ do not seem to be related to those generically at the source of the non-unitarity of the fractional-$N$ $O(N)$ models according to [3] (see the evanescent operator (7.82) of [3], which was first noticed in [50]).

## VII.   CONCLUSION AND OUTLOOK

We have shown in this work that the bootstrap of the $O(N)$ model was insensitive to the non-unitary nature of the model for both fractional $d > 3$ and fractional $N > 1$. In the process, we gave bootstrap and $\epsilon$-expansion estimates of a substantial amount of CFT-data in the range $(d, N) \in [3, 4] \times [1, 3]$. We then studied in more detail the limit $N \to 1$, and obtained the clear disappearance of the $O(N)$ islands below $N = 1$ (see [51] for another case where sizeable non-unitarities lead to disagreement between bootstrap and field-theory results). We obtained these results using the newly developed navigator method, and devised a simple pathfollowing algorithm which enabled us to sail from island to island efficiently. In some cases this led to an appreciable speedup of subsequent optimization runs, and in others it was shown to be necessary in finding the next island in the first place.

The disappearance of the $O(N)$ islands at $N_c = 1$ could be observed with the navigator method as the minimum of the navigator $\mathcal{N}(x)$ went above 0 at roughly $N_c$. This constitutes a much clearer signature of the disappearance of the island than could be possible with the usual binary bootstrap, where one could still wonder if some small island could have evaded the scan. As was already discussed in [1], we believe that the navigator method could be helpful in studying other systems where the behavior of some family of CFTs is expected to change at some critical value(s)

of the external parameters. This is the case for example for the $O(N)$ models near $(d,N) = (2,2)$, where a critical line $(d_c, N_c)$ (the "Cardy-Hamber" line) emerges from $(d,N) = (2,2)$ along which two fixed points collide [52, 53]. Some other examples would include the merger and annihilation of the critical and tricritical $q$-state Potts model along another critical line $(d_c, q_c)$ [54, 55], and the controversial fate of the $O(N) \times O(2)$ universality class supposedly describing phase transitions in certain classes of frustrated magnets, where there should also exist a critical line $(d_c, N_c)$ along which there is the merger and annihilation of the so-called "chiral" and "antichiral" fixed points [56] (see [6, 57] for previous bootstrap work). In cases like the $O(N) \times O(2)$ model, where the crossing equations involve lots of internal channels, we expect that the navigator will enable us to scan over many more internal exchanged operators than were considered in the past. Paired with the pathfollowing prescription laid out in this paper which should help sail in the $(d,N)$ plane more efficiently, this should lead to a better determination of the critical line $(d_c, N_c)$.

## ACKNOWLEDGEMENTS

The author thanks S. Rychkov for his many helpful comments in the elaboration of this paper. The author is also thankful to M. Paulos, B. van Rees, D. Simmons-Duffin, J. Henriksson and M. Reehorst for useful discussions, to Jiaxin Qiao and Junchen Rong for comments on the final draft, and to N. Su for all the help provided in setting up the numerics. The author is supported by a *Fonds de Recherche du Québec – Nature et technologies* B2X Doctoral scholarship, by Mitsubishi Heavy Industries (through an MHI-ENS Chair) and by the Simons Collaboration on the Nonperturbative Bootstrap. The computations in this work were performed on the Caltech High Performance Cluster, partially supported by a grant from the Gordon and Betty Moore Foundation.

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
