# Peer review of "Navigating through the O(N) archipelago"

_SciPost Physics_

## Round 1 · Referee Report · Andreas Stergiou (Referee 1) · 2022-5-16

Strengths

1- Clear exposition 2- Well thought out discussion of technical results

Weaknesses

1- Familiarity with the paper SciPost Phys. 11 (2021) 072 (arXiv:2104.09518) is largely assumed. (One has to recognize, however, that this is unavoidable to some extent.)

Report

It is by now well-established that nonunitarities arise in the $O(N)$ models when the spacetime dimension $d$ and/or $N$ become fractional, e.g. when $d$ is taken somewhere between 3 and 4. In this manuscript the author performs a systematic numerical bootstrap study of the $O(N)$ models in a certain range of $d$ and $N$ using the navigator method, which is a method co-developed by the author in previous work. Numerical bootstrap islands, which contain the $O(N)$ models, are followed in the $(d, N)$ plane, with the motivation of elucidating the type and degree of nonunitarity that the numerical conformal bootstrap can discern.

Older work has shown that nonunitarities associated with fractional $d$ or $N$ (when $N>1$) are not easily detectable by the numerical conformal bootstrap. This manuscript shows that, unfortunately, the navigator method suffers the same limitation. This is perhaps disappointing, but nevertheless it is an important result. A more positive result is that the navigator method is indeed able to diagnose nonunitarities encountered in the $O(N)$ models for $N<1$. These latter nonunitarities are rather large, as they are associated with squared OPE coefficients of low-lying operators turning negative. Taken together, the two sets of results presented by the author contribute to a clearer understanding of the degrees of nonunitarity that the most advanced numerical bootstrap tools can detect.

The manuscript is well-written and constitutes a valuable contribution to the numerical bootstrap literature. I recommend publication to SciPost Physics.

---

## Round 1 · Referee Report · Anonymous (Referee 2) · 2022-7-20

Strengths

1) Very thorough description of the technical analysis.

2) Highly relevant physical questions.

Weaknesses

1) The presentation combines technical details and physical results, making it difficult for non specialized audience to understand and appreciate the physics outcome.

Report

The paper applies a method previously developed by the same author and collaborators [SciPost Phys. 11 (2021) 072] to O(N) symmetric models, computing the scaling dimension of the relevant and sub-leading operators at the critical point as a function of the two parameters (d,N) i.e. the dimension and the symmetry index of the model.

At the beginning the paper is clearly presented and the two main scopes are delineated, namely:

1) Study the consequences of the non-unitary nature of the CFT due to fractional values of d and N.

2) Optimize the flow in the external parameter space and in the search space of conformal bootstrap.

The second aim is purely methodological and, therefore, naturally directed to the narrow community of conformal bootstrap. Nevertheless, even with this methodological goal in mind, I found the paper way too technical. In particular, after starting with well delineated aims and a broad discussion in Sec. I and II, Sec. III immediately becomes very technical and difficult to read for anyone who is not an expert in the method at hand.

Some general physics perspective is recovered only in Sec. V, but even there the physical results are intertwined with such a large amount of technical and numerical details that it is difficult to discern what's the real message of the paper.

I understand that the author is an expert of conformal bootstrap and desires to share with the reader his deep understanding of the methodology, but most of the audience will not be interested and probably miss the main message of the paper once confronted with a similar amount of technical information. The paper will be better served by a long appendix, where technical details and developments are worked out, while the main text should have been left to discuss the physics.

I understand that at this stage such a huge restyling of the article may be complicated, therefore I would advise the author to at least try to better separate the methodological findings, which concern aim (2) from the discussion of the non-unitary nature of the CFT, i.e. aim (1), which may be interesting for a broader physics community.

Despite its specialized nature, the paper may contain enough novel information to deserve publication on SciPost, provided that this and the following changes are made.

Requested changes

1) The author shall make an effort to guide the reader through the results by indicating which sections and subsections are of more technical nature (and can therefore be skipped by the uninterested reader).

2) The physical question of the non-unitary nature of the O(N) CFT needs to be put in better context. At present the author just refers to previous literature, mainly Refs [2,3], but it is not clear the relation between the picture presented in those studies and the findings of the present paper, in particular the following questions arise:

-) Was the fact that non-unitary contributions only arise at high energy already expected in Refs. [2,3] ? Or is this an unexpected result of the present paper?

-) Is it possible to make a bound on the smallness of non-unitary contributions and/or on the energy/scaling dimension threshold where they would become visible?

-) Non-unitary contributions in fractional $d$ are related to evanescent operators, but the source of non-unitarity at N-->1 is of a different nature (as noticed by the author). Can we say something on the reason why such evanescent contributions are not visible in the current approach? Is this a drawback of the methodology or is really the contributions from evanescent operators so small that will be not visible also with other approaches?

3) On page 4 the author writes: "We will resum the [$\epsilon$-expansion ] series using the algorithm of Borel-Leroy transform with conformal mapping laid out in Section V of [18].", but no further details are given, I understand that the procedure is complex but at least a brief outline shall be given in order for the readers to understand how are the curves in Fig.12 obtained. Especially, I would like to be clarified how the uncertainties displayed in Fig. 12 have been computed.

4) Finally, looking at figure 12, it is striking that the uncertainty of the bootstrap result is quite large with respect to the one of $\epsilon$-expansion, can this feature be improved in the future? How long has the computation of each of the single points in Fig. 12 run on the cluster? Are the present uncertainties competitive with the ones obtained with more conventional bootstrap approaches like the one described in Ref. [4]?

---

## Round 1 · Referee Report · Anonymous (Referee 3) · 2022-7-26

Strengths

1- The author made an effort to establish the validity of the method, with numerous crosschecks and addressing potential issues. 2-Carefully explains the assumptions, the limitations and potentialities of the method

Weaknesses

1-The paper is very technical and assumes a deep knowledge of the field and of the previous work of the author.
2-Figures are hard to read in the printed version, can only be visualised online.

Report

The manuscript represents an application and extension of the navigator method introduced in a previous by the same author. In this work the author applies the method to find approximate solutions of crossing equations of O(N) -model between 2 and 4 dimension by moving from one solution to another instead of scanning the parameter space.

The method is computationally more efficient and corroborates the expectation that CFTs with different values of N and spacetime dimensions are connected.

The pathfollowing algorithm is well explained, but it is largely assumed that the reader is familiar with the navigator method introduced in the previous paper.

The discussion about disappearance of solution for N<1 is very nice and convincing.

Requested changes

1-The author mentions several times that theories with non-integer values of d and N are not unitary but the algorithm is insensitive to this. It would be nice to establish this quantitatively, or at least give a more robust argument of why this is not the case. 2- Improve the quality of the figures, especially labels 3- Give additional details about the epsilon-expansion resummation

---

## Editorial Decision

resubmitted